



# Characteristics and Evolution of Brown Carbon in Western United States Wildfires

Linghan Zeng[1], Jack Dibb[2], Eric Scheuer[2], Joseph M. Katich[3,4], Joshua P. Schwarz[4], Ilann Bourgeois[3,4], Jeff Peischl[3,4], Tom Ryerson[3,4,a], Carsten Warneke[4], Anne E. Perring[5], Glenn S. Diskin[6], Joshua P. DiGangi[6], John B. Nowak[6], Richard H. Moore[6], Elizabeth B. Wiggins[6], Demetrios Pagonis[3,7,b], Hongyu Guo[3,7], Pedro Campuzano-Jost[3,7], Jose L. Jimenez[3,7], Lu Xu[8,c], Rodney J. Weber[1]

[1]Earth and Atmospheric Sciences, Georgia Institute of Technology, Atlanta, GA, USA
[2]College of Engineering and Physical Sciences, University of New Hampshire, Durham, NH, USA
[3]Cooperative Institute for Research in Environmental Sciences, University of Colorado Boulder, Boulder, CO, USA
[4]Chemical Sciences Laboratory, National Oceanic and Atmospheric Administration, Boulder, CO, USA
[5]Department of Chemistry, Colgate University, Hamilton, NY, USA
[6]NASA Langley Research Center, Hampton, VA, USA
[7]Department of Chemistry, University of Colorado Boulder, Boulder, CO, USA
[8]Division of Geological and Planetary Sciences, California Institute of Technology, Pasadena, CA, USA

[a]Now at: Scientific Aviation, Boulder, CO, USA
[b]Now at: Department of Chemistry and Biochemistry, Weber State University, Ogden, UT, USA
[c]Now at: Chemical Sciences Laboratory, National Oceanic and Atmospheric Administration, Boulder, CO, USA and Cooperative Institute for Research in Environmental Sciences, University of Colorado Boulder, Boulder, CO, USA

*Correspondence to*: Rodney J. Weber (rweber@eas.gatech.edu)

## Abstract

Brown carbon (BrC) associated with aerosol particles in western United States wildfires was measured between Jul. and Aug. 2019 onboard the NASA DC-8 research aircraft during the Fire Influence on Regional to Global Environments and Air Quality (FIREX-AQ) study. Two BrC measurement methods are investigated; highly spectrally-resolved light absorption in solvent (water and methanol) extracts of particles collected on filters and in-situ bulk aerosol particle light absorption measured at three wavelengths (405, 532, 664nm) with a photo acoustic spectrometer (PAS). A light absorption closure analysis for wavelengths between 300 and 700 nm was performed. The combined light absorption of particle pure black carbon material, including enhancements due to internally mixed materials, plus soluble BrC and a Mie-predicted factor for conversion of soluble BrC to aerosol particle BrC, was compared to absorption spectra from a power law fit to the three PAS wavelengths. For the various parameters used, at a wavelength of roughly 400 nm they agreed, at lower wavelengths the individual component-predicted particle light absorption significantly exceeded the PAS and at higher wavelengths the PAS absorption was consistently higher, but more variable. Limitations with extrapolation of PAS data to wavelengths below 405 nm and missing BrC species of low solubility that more strongly absorb at higher wavelengths may account for the differences. Based





on measurements closest to fires, the emission ratio of PAS measured BrC at 405 nm relative to carbon monoxide (CO) was on average 0.13 Mm$^{-1}$ ppbv$^{-1}$, emission ratios for soluble BrC are also provided. As the smoke moved away from the burning regions the evolution over time of BrC was observed to be highly complex; BrC enhancement, depletion, or constant levels with age were all observed in the first 8 hours after emission in different plumes. Within 8 hours following emissions, 4-nitrocatechol, a well characterized BrC chromophore commonly found in smoke particles, was largely depleted relative to the bulk BrC. In a descending plume where temperature increased by 15 K, 4-nitrocatechol dropped possibly due to temperature-driven evaporation, but bulk BrC remained largely unchanged. Evidence was found for reactions with ozone, or related species, as a pathway for secondary formation of BrC under both low and high oxides of nitrogen (NO$_x$) conditions, while BrC was also observed to be bleached in regions of higher ozone and low NO$_x$, consistent with complex behaviors of BrC observed in laboratory studies. Although the evolution of smoke in the first hours following emission is highly variable, a limited number of measurements of more aged smoke (15 to 30 hours) indicate a net loss of BrC. It is yet to be determined how the near-field BrC evolution in smoke affects the characteristics of smoke over longer time and spatial scales, where its environmental impacts are likely to be greater.

## 1. Introduction

Open biomass burning, which includes wildfires and prescribed burning, emits trace gases and aerosol particles into the atmosphere (Andreae, 2019). In the US, wildfires account for large burned areas (Kolden, 2019) and are increasing in frequency, especially in western regions (Burke et al., 2021; Mcclure and Jaffe, 2018). While wildfires can be beneficial to certain ecosystems (Thompson et al., 2011), aerosol particles produced from wildfires pose a substantial health threat (Akimoto, 2003; Regalado et al., 2006; Laumbach and Kipen, 2012; Fang et al., 2016; Chen et al., 2017); wildfire smoke may be more toxic than other sources of aerosol particles in terms of adverse respiratory impacts (Aguilera et al., 2021) and exposure can increase susceptibility to other respiratory hazards (Zhou et al., 2021). Smoke aerosol particles also produce observable optical effects and influence the planetary radiation balance (Zhang et al., 2020a). However, wildfire smoke impacts are highly complex. Following emission, both the toxicity and optical properties substantially change as the particles undergo atmospheric processing (Forrister et al., 2015; Wong et al., 2019b; Kleinman et al., 2020; Leblanc et al., 2020). Because the atmospheric lifetime of fine aerosol particles can range from about 5 to 30 days (Williams et al., 2002; Kristiansen et al., 2016), wildfire particles can have substantial environmental impacts over local, regional, and global scales (O'dell et al., 2021).

By mass, particles emitted from wildfires are mainly carbonaceous, such as organic aerosol (OA) and black carbon (BC) species (Andreae, 2019; Garofalo et al., 2019). Unlike most OA that predominantly scatters light (i.e., the imaginary part of the complex refractive index is near zero), a small mass fraction of OA absorbs light. For these species, the spectral light absorption is characterized by increasing absorption with decreasing wavelength, resulting in a yellow or brown appearance,



and is hence referred to as brown carbon (BrC) (Andreae and Gelencsér, 2006). Globally, biomass burning is likely the predominant source of BrC (Zeng et al., 2020), with lesser contributions from incomplete combustion of bio- (Saleh et al.,

2015; Lei et al., 2018) and fossil-fuels (Healy et al., 2015; Olson et al., 2015). BrC is chemically complex and, unlike BC, unstable. Saleh (2020) has proposed a framework to help reduce this complexity by grouping BrC into four broad categories that lie on a continuum from very weakly absorbing (VW-BrC), through weakly absorbing (W-BrC) and moderately absorbing (M-BrC), then up to strongly absorbing (S-BrC) where the BrC has optical and physical properties approaching those of BC (Adler et al., 2019; Cheng et al., 2021) (see Figure 3). These classifications separate BrC by sources and characteristics such

as molecular weight, volatility, and solubility. By this method, all the characteristics are delineated by the BrC light absorption wavelength dependence (Absorption Angstrom Exponent; AAE) and mass absorption cross-section (MAC) or the imaginary part of the complex component ($k$) of the refractive index at a specific wavelength (e.g., 405 nm or 550 nm).

Characterizing BrC can provide insight into the environmental effects of wildfire emissions. Estimates from early model

simulations suggest that BrC is a non-negligible warming agent (Feng et al., 2013; Saleh et al., 2015; Wang et al., 2018). Pole-to-pole BrC measurements through the Atlantic and Pacific Basins showed that for the regions where measurements were made, BrC contributed in the range of 7 to 48% to the top of atmosphere direct radiative effect (DRE) relative to all light-absorbing carbonaceous particles (BC+BrC), and that most of the BrC was from biomass burning emissions transported over long distances (> 1,000's of km) (Zeng et al., 2020). Measurements have also shown that the prevalence of BrC relative to BC

increases in the atmospheric column with increasing altitude, especially in the range of about 5 to 13 km (Liu et al., 2014), possibly due to differences in cloud processing of BrC versus BC (Zhang et al., 2017). A global simulation including differences in atmospheric column BrC and BC distributions predicted that BrC, largely from biomass burning, accounted for more than 25% of the DRE compared to BC globally, and atmospheric heating in the tropical mid- and upper-troposphere due to BrC was larger than BC (Zhang et al., 2020a). BrC may also reduce the ultraviolet actinic flux sufficiently to affect

atmospheric photochemical reactions (Jo et al., 2016; Mok et al., 2016; Dasari et al., 2019). In terms of toxicity, BrC has been found to often correlate with aerosol oxidative potential (Verma et al., 2015), which has been linked to adverse cardiorespiratory effects (Bates et al., 2019). By slowing the photochemical aging processes of pollutants, such as heavy metals or other organic compounds, BrC could increase the dispersion of co-emitted carcinogenic compounds (Shrivastava et al., 2017).

Molecular-level characterization of BrC particles provides insights into their optical properties, formation and scavenging mechanisms, and toxicity. In early studies, nitroaromatic compounds were identified as BrC chromophores in particles from incomplete combustion, including biomass burning emissions (Claeys et al., 2012; Lin et al., 2016). Nitroaromatic species both absorb light and are known to be highly toxic (Bandowe and Meusel, 2017; Tian et al., 2020). Zhang et al. (2013) reported

00   eight nitro-aromatic chromophores in urban ambient aerosols accounting for only ~4% of the light absorption at 365 nm wavelength, whereas Desyaterik et al. (2013) found that these same compounds comprised approximately 50% of BrC in cloud



water samples influenced by agricultural burning events. 4-nitrophenol, 4-nitrocatechol, and their derivatives are now commonly identified BrC species (Bluvshtein et al., 2017; Hems and Abbatt, 2018); other identified chromophores include a range of polycyclic aromatic hydrocarbons (PAHs) derivatives and polyphenols that span wide molecular weights and structures (Lin et al., 2016). Carbonyl functional groups are a common feature of BrC chromophores (Laskin et al., 2015; Lin et al., 2015a; De Haan et al., 2017). Evidence suggests that strongly absorbing chromophores comprise a small mass fraction of OA in biomass burning smoke, but dominate the overall BrC absorption (Nguyen et al., 2012; Laskin et al., 2014).

BrC is similar to the bulk OA in that it can be directly emitted (primary BrC). A fraction of BrC is semi-volatile (Jai Devi et al., 2016) and may be a component of the secondary organic aerosol (SOA). This secondary BrC can be formed from a range of species and processes, such as reactions between aromatic VOCs with ozone ($O_3$) or the hydroxyl (OH) or nitrate ($NO_3$) radicals (Lee et al., 2014; Jiang et al., 2019; Fan et al., 2020), aqueous reactions involving carbonyl function groups with ammonium sulfate or PAHs with illumination (Nguyen et al., 2012; Haynes et al., 2019), and heterogeneous reactions of isoprene on acidic particles (Limbeck et al., 2003).

Laboratory studies demonstrate that the behavior of freshly-formed BrC is highly complex, where soon after emission both photo-enhancement and photobleaching can occur. Table 1 provides a brief summary of processes that can affect BrC once emitted by fires. Here we highlight only a few studies amongst many, and a number of review articles provide more details (Moise et al., 2015; Laskin et al., 2015; Yan et al., 2018). Zhong and Jang (2014) tracked the light absorption coefficient of ambient aerosols from biomass burning smoke captured in an outdoor smog chamber (with exposure to ambient light), finding the BrC mass absorption coefficient increased in the morning and gradually decreased thereafter. Similar behaviors were observed for aqueous phase BrC from laboratory-generated biomass burning aerosols exposed to UV light and reactions with the OH radical (Zhao et al., 2015; Wong et al., 2017; Wong et al., 2019a). Functionalization of nitrophenol molecules through oxidation by aqueous OH radicals and fragmentation of aromatic structures to smaller oxygenated molecules by direct photolysis was observed to first produce a photo enhancement, followed by photobleaching (Hems and Abbatt, 2018). Oxidation by $O_3$ has been observed to bleach BrC (Sareen et al., 2013; Fan et al., 2020), but BrC absorption can also increase at the beginning of the $O_3$ oxidation process (Kuang and Shang, 2020). The chemical processing of individual chromophores has also been studied. The light absorption of 4-nitrocatechol exhibited a wavelength-dependent change; a decrease in absorption between wavelengths of 300 and 380 nm and an increase in absorption below 300 nm or above 380 nm with increasing illumination time (Zhao et al., 2015). Reactions with the $NO_3$ radical produced nitrated organics, such as nitroaromatics, that contributed to aerosol particle BrC (Bluvshtein et al., 2017; Lin et al., 2017; Jiang et al., 2019; Li et al., 2020; Mayorga et al., 2021).

The atmospheric fate of BrC is not well understood, yet this determines its environmental impacts. In laboratory studies, time scales for significant bleaching of secondary BrC are on the order of minutes to several hours to days (Bluvshtein et al., 2017;





Lin et al., 2017; Jiang et al., 2019; Li et al., 2020; Mayorga et al., 2021), but atmospheric observations from wildfires show more complex behaviors. For relatively fresh smoke, Palm et al. (2020) conclude that a balance between dilution-driven evaporation of primary wildfire smoke chromophores and formation of secondary BrC led to the observation in nine fire plumes of a near-constant level of light absorption by BrC for smoke up to at least 6 hours old. Wu et al. (2021) found that

smoke from West-African prescribed fires that started with minor levels of BrC, but were rich in BC, had continual increases in BrC with plume age for up to 12 hours. Studies tracking smoke over longer time scales have shown an overall loss of BrC, with a BrC characteristic lifetime (e.g., e-folding lifetime) ranging from ~10 hours to days (Forrister et al., 2015; Wang et al., 2016). Some fraction of BrC is very resistant to losses, allowing it to persist and become widely dispersed (Kieber et al., 2006; Hecobian et al., 2010; Liu et al., 2015a; Washenfelder et al., 2015; Selimovic et al., 2020; Zeng et al., 2020). The longer-term

stability of BrC may depend largely on the molecular weight of the chromophores. Laboratory studies show that low molecular weight chromophores tend to be rapidly bleached, whereas high molecular weight BrC species were more recalcitrant and their relative contribution to overall BrC increased as the particles aged (Di Lorenzo et al., 2017; Wong et al., 2017; Wong et al., 2019a). Overall, both field and laboratory studies show that the evolution of BrC from wildfire smoke is highly complex with many competing processes that may produce widely different smoke evolution behaviors in the regions relatively near the

fires.

To gain a better understanding of the emissions from wildfires and their evolution within the first hours, airborne measurements were conducted as a part of NASA/NOAA Fire Influence on Regional to Global Environments and Air Quality (FIREX-AQ). One of the main objectives was studying open biomass burning in the western US in the summer of 2019 (22 Jul. 2019 – 17

Aug. 2019). Here, we report mainly on the characteristics and evolution of BrC chromophores based on a sequential solvent (water, then methanol) extraction method with liquid spectrophotometric measurements and aerosol particle BrC inferred from a photoacoustic spectrometer (PAS).

## 2. Method

### 2.1. The aircraft campaign

The FIREX-AQ study of 2019 included measurements from the NASA DC-8 aircraft, two NOAA Twin Otter (FIREX-CHEM and FIREX-MET) aircraft, and additionally, two ground-based Mobile Laboratories. Broad details of the campaign implementation and payload, including the large suite of gas and particle instruments, are provided in the FIREX-AQ white paper (https://www.esrl.noaa.gov/csl/projects/firex-aq/whitepaper.pdf, last access: 21 Jan. 2022). In the following, we focus on data collected from the NASA DC-8 research aircraft in wildfire smoke.



## 2.2. Instrumentation

### 2.2.1. Light absorption measurements

Two methods were used to determine BrC in this study, an off-line filter-based approach and in situ measurements from a photoacoustic spectrometer (PAS). Following a number of past studies (Liu et al., 2014; Liu et al., 2015a; Zeng et al., 2020), the light absorption of soluble BrC species (an operationally defined parameter) was measured by a liquid-based spectrophotometric method on solvent extracts of particle-laden filters. The spectrometer (USB-4000, Ocean Optics, Dunedin, FL) was coupled with a Long-path Waveguide Capillary Cell (2.5 m optical path; LWCC-3250; World Precision Instruments, Sarasota, FL) and a broadband UV-VIS-NR light source (DH-mini; Ocean Optics, Dunedin, FL). A similar approach was used on a newly developed online mist chamber water-soluble particle collection system deployed for the first time in this study (Zeng et al., 2021). Detailed operating procedures and data processing are described elsewhere (Zeng et al., 2021). Here we only focus on the filter data since we are interested in more than just water-soluble BrC species. Briefly, aerosol particles with an aerodynamic diameter less than 4.1 µm were collected onto Teflon filters at intervals typically less than 5 mins for samples within smoke plumes and a maximum of ~20 mins for background air sampling. Filters were extracted at a later date sequentially, first with water, then after air drying, with methanol. In each case, the resulting liquid extract was filtered (0.22 µm pore size syringe filter) and then injected into the LWCC via a syringe pump. The absorption spectra over wavelengths from 300 to 700 nm at ~1 nm resolution was recorded relative to that of the pure solvent, resulting in the light absorption spectra of water-soluble (WS) and methanol-soluble (MS) chromophores of species in the ambient aerosol particles. Light absorption measured in the waveguide ($A_\lambda$) is converted to an ambient aerosol light absorption coefficient ($Abs_\lambda^{LWCC}$) using:

$$Abs_\lambda^{LWCC} = A_\lambda \frac{V_l}{V_a \cdot l} \ln(10) \qquad (1)$$

where $V_a$ is the volume of air that passed through the filter, $V_l$ is the volume of solvent used in the extraction, and $l$ is the LWCC optical path length (nominally 2.5 m). The total-soluble (TS) light absorption measurement is defined here as the sum of the light absorption coefficients at each wavelength measured from the WS and MS extracts ($Abs_{TS,\lambda}^{LWCC} = Abs_{WS,\lambda}^{LWCC} + Abs_{MS,\lambda}^{LWCC}$), as sequential extraction has been shown to be comparable to methanol extraction alone (Liu et al., 2015a). (Note that water then methanol measurements of BrC were done to provide both WS and total BrC data, since some instruments are only capable of WS BrC measurements, e.g., the online systems (Zeng at al., 2021)). As will be discussed, this extraction does not necessarily account for all of the BrC, since some chromophores may not be soluble in these solvents. As in past studies, for simplicity, overall BrC levels often characterized by light absorption at one wavelength, such as for the solvent method at 365 nm for both WS BrC and TS BrC ($Abs_{WS,365nm}^{LWCC}$ and $Abs_{TS,365nm}^{LWCC}$). Light absorption data over the whole spectrum are available from the NASA data archive (FIREX-AQ 2019; https://doi.org/10.5067/suborbital/firexaq2019/data001, last access 21 Jan. 2022). For this method, the limits of detection (LOD) were determined by three standard deviations of the blank measurements and are 0.10 Mm$^{-1}$ and 0.26 Mm$^{-1}$ for WS and MS BrC, respectively, at 365 nm. At other wavelengths from



300 nm to 700 nm, the LODs ranges between $0.08 - 0.52$ Mm$^{-1}$ for both $Abs_{WS,\lambda}^{LWCC}$ and $Abs_{MS,\lambda}^{LWCC}$. The uncertainties associated with the absorption measurements were calculated by propagating the uncertainties from sampling (air flow rates and sampling time), filter extraction, and the absorption measurement, and are estimated to be 16% for WS BrC, 19% for MS BrC and 25% for TS BrC at 365 nm. The uncertainties are larger towards higher wavelength because the measured absorption is closer to the blank measurement; the uncertainties are ~60% for $Abs_{WS,700nm}^{LWCC}$ and $Abs_{MS,700nm}^{LWCC}$, and therefore the uncertainty for $Abs_{TS,700nm}^{LWCC}$ can exceed 85%. There are several advantages of the BrC measurement based on aerosol extracts: (1) the majority of insoluble absorbers, for example BC or mineral dust particles, were filtered out (Zeng et al., 2021) making it a direct measurement of BrC; (2) the light absorption can be measured over a broad wavelength spanning from UV to visible range at high spectra resolution (300 nm – 700 nm, at ~1 nm resolution). However, aerosol particle size and morphology information are lost, so light absorption ($Abs_{\lambda}^{LWCC}$) measured in the LWCC is not directly comparable to results from aerosol optical instruments since it does not consider particle size and other related effects. In the following, we denote the absorption coefficients for just chromophores in liquids by $Abs_{\lambda}^{LWCC}$ and the estimated coefficients for the chromophores in aerosol particles by $b_{ap,BrC,\lambda}$. A further limitation is that a fraction of non-polar chromophores may not be extracted efficiently in water or organic solvents (Corbin et al., 2019; Shetty et al., 2019).

A photoacoustic spectrometer (PAS) was deployed on the DC-8 providing real-time measurements of dry aerosol absorption of fine particles (diameters < 2.5 µm) at three wavelengths: 405 nm, 532 nm, and 664 nm (Lack et al., 2012; Langridge et al., 2013). In this instrument, the light at a specific wavelength absorbed by an aerosol particle is converted to an acoustic pressure wave that is detected with a microphone. Uncertainties for these data are estimated to be 20%, mainly from calibration, pressure variation and optical alignment issues (Langridge et al., 2011). Note that the PAS does not directly measure BrC absorption, it must be calculated as the difference between total aerosol light absorption and light absorption by BC (here BC is referred to as the overall light absorption properties of BC, which includes the absorption by BC and any absorption enhancement due to coatings). Since BC, in most cases, dominates the light absorption at all wavelengths, BrC absorption inferred by optical instruments like the PAS can have a large uncertainty due to BrC being calculated by subtraction of two similar magnitude numbers. In smoldering smoke plumes this is less an issue due to the high levels of BrC relative to BC. As the PAS only provided data at a limited number of wavelengths, mostly in the visible wavelength range, extrapolation from visible to UV wavelengths is required, which also leads to further uncertainties when using these data to infer optical properties over a broad spectral range (Liu et al., 2015c).

### 2.2.2. Other measurements

Refractory black carbon (rBC) mass concentration was measured by a single particle soot photometer (SP2), which quantified rBC mass in individual particles in the 0.090 to 0.550 µm size range (volumetric-equivalent diameter assuming 1.8 g cm$^{-3}$ void-free density) based on the incandescence signal they generated when passing through a laser beam (Stephens et al., 2003;



Schwarz et al., 2006). The amplitude of the BC incandescence signal is related to the amount of refractory material contained in the illuminated particle. Integrated BC concentrations have been adjusted to account for accumulation-mode BC outside of the SP2's detection size range (Schwarz et al., 2008) on a per-flight basis. Use of a dilution system to reduce particle loads to single-particle instruments on the DC-8, increased total SP2 uncertainty to larger values than typical. Total uncertainty was estimated to be 40% in integrated rBC mass mixing ratios in its size-range with the dilution system.

OA mass concentration were measured for particles nominally up to 1 µm diameter by a high-resolution time-of-flight Aerodyne aerosol mass spectrometer (HR-ToF-AMS) (Decarlo et al., 2008; Decarlo et al., 2006). The uncertainty of the OA mass concentration was estimated to be 38% (2σ), dominated by uncertainty in particle collection efficiency due to particle bounce, and absolute and relative ionization efficiency (Bahreini et al., 2009). A collocated extractive electrospray ionization time-of-flight mass spectrometer (EESI-ToF-MS) (Lopez-Hilfiker et al., 2019; Pagonis et al., 2021), measured the mass concentration of 4-nitrocatechol (4-NC). The instrument was limited to pressure altitudes below 7 km above sea level, and the measurement uncertainty was estimated to be 47% (2σ). HR-ToF-AMS and EESI-ToF-MS shared a National Center for Atmospheric Research (NCAR) High-Performance Instrumented Airborne Platform for Environmental Research Modular Inlet (HIMIL) (Stith et al., 2009), together with high efficiency particulate air (HEPA) filter for background measurements and a calibration system (Pagonis et al., 2021).

Aerosol number size distribution was measured by a laser aerosol spectrometer (LAS, model 3340, TSI Incorporated, Shoreview, MN), operated behind a monotube Nafion dryer (Moore et al., 2021). The reported size range is from ~100 nm to 4 µm (Brock et al., 2019). LAS was size-calibrated with mobility-classified ammonium sulfate particles. The LAS uncertainty is estimated to be 20% to account for variability in smoke aerosol refractive index.

Carbon monoxide (CO) mixing ratios were measured by a diode laser spectrometer system, referred to by its historical name Differential Absorption Carbon Monoxide Measurements (DACOM) with a measurement uncertainty of 2 ppbv (Warner et al., 2010; Sachse et al., 1991). $O_3$ and $NO_x$ were measured with the NOAA Nitrogen Oxides and Ozone ($NOyO_3$) 4-channel chemiluminescence instrument with measurement uncertainty of 5-10 pptv ± 3% and 9%, respectively (Pollack et al., 2010; Bourgeois et al., 2022). All the data presented here are at standard temperature and pressure (273K and 1013 mbar). High-resolution 1 Hz data (e.g., measurements by PAS, BC, CO, $O_3$, $NO_x$, OA, and 4-NC) were merged to low time resolution data (i.e., 10s data or filter sampling interval) depending on the analysis performed.

**2.3 Calculation of light absorption coefficients**

Various aerosol particle light absorption coefficients as a function of wavelength are determined from these measurements and compared. From the PAS and SP2 measurements, light absorption due to just BrC ($b_{ap,PASBrC,\lambda}$) is obtained from the difference of the measured (total) absorption ($b_{ap,PAS,\lambda}$) and the absorption by rBC ($b_{ap,rBC,\lambda}$) as a function of wavelength by:


$$b_{ap,PASBrC,\lambda} = b_{ap,PAS,\lambda} - b_{ap,BC,\lambda} = b_{ap,PAS,\lambda} - E_\lambda \cdot b_{ap,rBC,\lambda} \qquad (2)$$

where $b_{ap,BC,\lambda}$ is the overall light absorption coefficient of BC, including a lensing effect (enhancement, $E_\lambda$) due to

coatings on rBC. We estimate the light absorption coefficient of rBC from the published properties of pure BC by:

$$b_{ap,rBC,\lambda} = c_{rBC} \cdot MAC_{rBC,550nm} \cdot \left(\frac{\lambda}{550nm}\right)^{-AAE_{rBC}} \qquad (3)$$

where in Eqn (3) the absorption coefficient of rBC ($b_{ap,rBC,\lambda}$) is estimated from the SP2-measured refractory BC mass

concentration ($c_{rBC}$), with an assumed rBC AAE of 1 and mass absorption cross-section ($MAC_{rBC,550nm}$) of pure BC of $7.5 \pm$

1.2 m$^2$ g$^{-1}$ at 550 nm (Bond and Bergstrom, 2006). The AAE of rBC can range from 0.8 to 1.4, and a clear coating does not

alter the AAE of rBC significantly (Lack and Langridge, 2013). However, rBC heavily coated with chromophores can result

in an AAE of 3 (Zhang et al., 2020b). The enhancement factor ($E_\lambda$) in the rBC absorption due to coatings is not well known as

it depends on particle characteristics that are most often not fully measured, such as the coating or BC geometry (Lack and

Cappa, 2010; Luo et al., 2018) and the coating optical properties (e.g., clear or absorbing) (Liu et al., 2017; Wu et al., 2018;

Zhang et al., 2018). The absorption enhancement has been observed to be less than 5% or as high as 250%, corresponding to

$E_\lambda$ of 1–3.5, and the enhancement effect is larger towards lower wavelengths (Zeng et al., 2020; Zhang et al., 2017). We assume

the AAE of rBC ($AAE_{rBC}$) is 1, as in other studies (Zeng et al., 2020; Zhang et al., 2017). We also use a constant $E_\lambda$ of 1.6 at

all wavelengths, which is a typical level reported (Wu et al., 2018; Fierce et al., 2020), and consistent with the $E_\lambda$

parameterization of Chakrabarty and Heinson (2018) based on our coating levels observed in one smoke plume (average ratio

of rBC plus coating mass to rBC mass of approximately 4.5).

We compare the overall PAS-measured light absorption coefficient to an overall light absorption coefficient calculated from

the contributions of individual carbonaceous light-absorbing components. This predicted light absorption coefficient as a

function of wavelength ($b_{ap,predicted,\lambda}$) is determined by:

$$b_{ap,predicted,\lambda} = E_\lambda \, b_{ap,rBC,\lambda} + b_{ap,TSBrC,\lambda} = E_\lambda \cdot b_{ap,rBC,\lambda} + K_\lambda \cdot Abs_{TS,\lambda}^{LWCC} \qquad (4)$$

which is the sum of the light absorption by pure rBC with an added lensing effect, plus BrC measured in solution (TS BrC,

i.e., the sum of all chromophores in the extraction solvent, $Abs_{TS,\lambda}^{LWCC}$) and converted to BrC light absorption ($b_{ap,TSBrC,\lambda}$) by

aerosol particles by multiplying by a factor, $K_\lambda$. Several studies have used Mie theory to calculate the conversion factor ($K_\lambda$)

at a specific wavelength, typically 365 nm (Liu et al., 2013; Washenfelder et al., 2015; Shetty et al., 2019). Reported $K_{365nm}$

values are in the range of 1.8 to 2.3. Here we perform a more detailed Mie theory calculation and determine $K_\lambda$ over the liquid

spectrophotometer-measured wavelength range of 300 to 700 nm.

## 2.4 Mie theory calculation to convert solution to particle light absorption coefficients

Mie theory has been applied to convert the absorption coefficient from soluble BrC ($Abs_\lambda^{LWCC}$) in liquid extracts to an ambient

aerosol absorption coefficient ($b_{ap,BrC,\lambda}$) from the soluble chromophores. We do this with a conversion factor of $K_\lambda$, which is

determined based on a number of assumptions, including: (1) The BrC-containing aerosol particles are spherical; (2) BrC





chromophores are uniformly distributed through the particle; (3) the size distribution of BrC is the same as that of the OA aerosols (since in intense smoke plumes most of the aerosol is composed of organic species, this is a good assumption); and (4) the BrC aerosol is externally mixed with other light absorbers (BC), since BrC is only a small fraction of OA, most BC coating is likely to be non-absorbing OA species and the majority of BrC part of the OA not containing BC. Aerosol size distribution measured by the LAS (particle diameters between ~ 100 nm and 4.8 μm) were fitted with a log-normal distribution to account for data out of the size range (i.e., particles smaller than 100 nm). The LAS number distribution was scaled by the AMS OA mass concentration to estimate the number distribution of just OA. This was done by calculating the mass distribution from the LAS number distribution, by assuming spherical particles of density 1.4 g cm$^{-3}$, times the scaling factor, and integrating over the size range of zero to 1 μm. The scaling factor was adjusted so that the integrated LAS mass distribution equaled the AMS OA. MAC values were calculated based on the AMS OA mass ($Abs_{TS,\lambda}^{LWCC}/OA$). Detailed procedures for Mie theory calculations can be found elsewhere (Liu et al., 2013). We also assume in the Mie calculations a particle density of 1.4 g cm$^{-3}$ and the real part of the particle refractive index ($n$) of 1.55.

## 2.5 Age of smoke plumes

The age of species in a smoke plume advected from a fire can be estimated in a number of ways, but are only rough estimates due to the large spatial scales of the fires investigated, and other factors. Chemical age can be estimated based on known differential reaction rates of species, or the physical age can be estimated. Here we estimate the physical age based on air mass trajectories comprising two components: advection age and plume rise age (Liao et al., 2021). The advection time was estimated by the HYbrid Single-Particle Lagrangian Integrated Trajectory (HYSPLIT; (Rolph et al., 2017; Stein et al., 2015)) from the DC-8 aircraft location relative to the smoke source identified using the MODIS/ASTER airborne simulator (MASTER; (Hook et al., 2001)) with multiple high-resolution meteorological dataset, including High-Resolution Rapid Refresh (HRRR), North American Mesoscale Forecast System (NAM CONUS nest), and Global Forecast System (GFS). The plume rise time to the trajectory height was obtained from MASTER fire altitude by assuming a vertical velocity of the air mass (12 m s$^{-1}$ in pyroCb and 7 m s$^{-1}$ otherwise). The typical uncertainty of estimated plume age was approximately 27%.

## 2.6 Normalized Excess Mixing Ratio (NEMR)

A number of parameters are used to assess the evolution of species of interest in fire plumes after emission from the burning regions. The normalized excess mixing ratio (NEMR) is the ratio of the enhancement (in-plume minus out of plume, the latter being the background) in the species of interest to the enhancement of a long-lived co-emitted species, such as CO, or CO$_2$ (Hobbs et al., 2003; Garofalo et al., 2019). In the following analysis, we use CO as the conservative tracer as its lifetime is ~ 1 month and it is much more enhanced in the plume relative to the background than CO$_2$. The change of NEMR with plume age indicates the total gain or loss of the species of interest during the plume advection from the burning region by excluding the change in concentration just due to physical dilution. Due to large enhancements in BrC in the smoke plumes relative to the regional background, we assume the BrC (both soluble BrC and PAS BrC) outside the plume was zero, so the NEMR$_{BrC}$





can be estimated to be BrC/ΔCO. The NEMR of rBC (NEMR$_{rBC}$) is also used to assess whether there was any significant change in the burning conditions that affected aerosol particle concentrations when sampling at different downwind locations in the plume.

## 3. Results and Discussion

### 3.1 Fire plume sampling

The DC-8 aircraft conducted 23 individual flights during FIREX-AQ, including 13 flights characterizing wildfires in the western United States, 8 flights targeting prescribed burning plumes, and 2 transit flights. Here we focus on the wildfire smoke sampled. Among the 13 flights investigating wildfires, 404 filters were collected in the western US including 268 filters (~66%) fully or partially within fire plumes. The details of the various plumes are given in Table 2.

As an example of how smoke from a specific fire was investigated (Williams Flats fire, 7 Aug. 2019, UTC 23:34 to 24:46, Table 2), the flight path and time series of the CO mixing ratio and aircraft altitude (GPS altitude) are shown in Figure 1. For this flight, the DC-8 departed from the Boise Airport (BOI, ID) on 7 Aug. 2019 UTC and flew toward the northeast to trace the aged plumes coming from the Williams Flats fires, which had been forecasted by models and observed in satellite images. This smoke was detected as a subtle enhancement in CO between 22:00 – 23:00 UTC (Figure 1b). Then, the DC-8 aircraft maneuvered to approach the fresh Williams Flats smoke from the downwind side at altitudes between ~3800-5200 m above sea level. Two lawn-mowing patterns, which resulted in semi-lagrangian sampling of the main smoke plume, were made in the late afternoon local time with 19 individual plume transects; 10 in the first pattern during local time 16:34-17:46 and 9 transects in the second pattern during local time 18:15-19:12. For the filter sampling, although the sampling goal was to only collect particles when the DC-8 was in the smoke plume, an inevitable small amount of background air was also collected on the filter samples due to the fast-moving aircraft (typical speed 200 m s$^{-1}$) and because the exact edges of the plumes were ambiguous. In-plume sampling was identified by enhancements in concentrations of CO, CO$_2$, and rBC; for example, in most cases in-plume was characterized by a CO enhancement of at least 200 ppbv over the background CO concentration. Figure 1b shows that the plume could be readily identified by large increases in CO. Following this, the aircraft made an excursion to the southeast and then returned to the Boise Airport.

### 3.2 Overall characteristics of BrC in smoke

When sampling in the various wildfire smoke plumes, soluble BrC absorption, including WS BrC and TS BrC, and BrC inferred from the PAS were highly correlated with various other gas and aerosol species expected to be emitted by the fires, such as CO, BC, OA, hydrogen cyanide (HCN) and acetonitrile (CH$_3$CN), the latter two measured by a Proton-transfer-reaction mass spectrometry (PTR-MS) (see Table 3). The various measurements of BrC (WS, TS and PAS) were highly correlated amongst themselves, with the highest correlation between TS BrC and PAS BrC. Compared to WS and TS BrC, PAS BrC also





had higher correlations with rBC and gas phase smoke species, possibly due to limitations with the offline filter method, or that PAS BrC is a more comprehensive measurement of BrC (includes possibly insoluble BrC species missed in the filter-solvent extraction method).

### 3.2.1 Soluble BrC

For all smoke samples collected during FIREX-AQ, WS BrC ($Abs_{WS,365nm}^{LWCC}$) accounted for 45% ± 16% (mean ± standard deviation) of TS BrC at 365 nm ($Abs_{TS,365nm}^{LWCC}$) (Figure S1, TS vs WS BrC at 365 slope=2.23, intercept=0, $R^2$=0.91), which are similar to levels (~45%) observed in fresh biomass burning plumes during the DC3 campaign that also investigated summertime western US wildfires (Liu et al., 2015a). This fraction is slightly lower than the observation in highly aged biomass burning in the ATom study (smoke transported from the continents to remote marine regions), which was 53% ± 17%, and could be explained by the ATom BrC being more oxidized (more aged) with a higher hygroscopicity (Duplissy et al., 2011).

The spectral characteristics of BrC are often characterized by the Absorption Angstrom Exponent (AAE). To cover as many of the short wavelengths as possible, we calculated the AAE from the light absorption measured between 300 nm to 500 nm. (Note, that calculating the AAE by fitting data with a power law is dependent on the wavelength range utilized (Moosmüller et al., 2011)). Including all measurements in the identified smoke plumes of this study, the AAE for TS BrC (sum of the water and methanol extracts) was on average 4.2 ± 1.6 (mean ± stdev). This is lower than the AAE for just WS BrC (AAE=5.1 ± 1.3), and likely a result from less-polar chromophores that are extracted in methanol and not water absorbing more light in the higher wavelength range compared to water-soluble chromophores (Liu et al., 2015a; Zhang et al., 2013). The AAE of the overall absorbing aerosol determined from fitting the absorption at the 3 PAS wavelengths with a power law for these same smoke plumes was 1.49 ± 0.52 (mean ± stdev), while the BrC determined from the PAS had an AAE of 2.07±1.01 (mean ± stdev).

### 3.2.2 BrC emissions and classification

The emission ratio of BrC (ER$_{BrC}$) can be estimated from the various fires as the ratio of ΔBrC to ΔCO (i.e., NEMR$_{BrC}$), assuming there was little atmospheric processing between the emission and time of measurement. To estimate emissions, we only use BrC data from wildfires with transport time less than two hours. The results are shown in Figure 2. For PAS-predicted BrC at a wavelength of 405 nm (the lowest PAS measurement wavelength) the ER$_{BrC}$ (slope) was 0.131 ± 0.001 Mm$^{-1}$ ppbv$^{-1}$ (the ± is the fit uncertainty in the slope at a 1-sigma confidence interval). For just water-soluble BrC the ER$_{BrC}$ was 0.071 ± 0.003Mm$^{-1}$ ppbv$^{-1}$, and for TS BrC 0.163 ± 0.006 Mm$^{-1}$ ppbv$^{-1}$. We calculated the soluble BrC ERs at 365 nm since this is the wavelength most often used to characterize BrC using a single wavelength. These data are BrC in the solution and not corrected for conversion to ambient particle (factor $K_\lambda$, discussed below, is not applied here) since many studies report BrC measured in solution. (Note that the TS BrC is higher than PAS BrC because it given at a lower wavelength, see Figure S2).





The wildfire BrC optical properties can be mapped onto the classification proposed by Saleh (2020) to provide a rough characterization and test the approach as a parameterization. For the PAS and filter data we determined the BrC AAE and the mass absorption cross-section (MAC). For the PAS, the $MAC_{PASBrC,405nm}$ was determined from the ratio of $b_{ap,PASBrC,405nm}$ to OA mass measured by the AMS, and $AAE_{PASBrC}$ was calculated from the power law fit to $b_{ap,PASBrC,405nm}$, $b_{ap,PASBrC,532nm}$, and $b_{ap,PASBrC,664nm}$. PAS data are shown with the Saleh BrC characteristics identified by regions in the boxes. In Figure 3a, these wildfires data are best characterized as M-BrC and had little relation to the modified combustion efficiency (MCE). The MCE for samples from wildfires encountered in FIREX-AQ was around 0.9, which is at the boundary of smoldering and flaming burning conditions. However, the MCE did not span a wide range for the fires investigated and so there was little range to produce a clear separation for different burning conditions. Our smoke data also shows little correlation between BrC $AAE_{PASBrC}$ and $\log_{10}(MAC_{PASBrC,405nm})$. Figure 3b shows a similar plot for the TS BrC (without applying the conversion factor $K_\lambda$ to convert to aerosol absorption). Most of these data are outside of the Saleh's categorization, but they are shifted to the upper left relative to the PAS data, consistent with the idea that PAS BrC contains relatively more weakly absorbing species. Like the PAS BrC, TS BrC also does not show a clear trend with MCE, nor a correlation between AAE and $\log_{10}(MAC)$.

### 3.3. Comparing methods for determining BrC

A closure analysis is performed to compare the PAS and solution methods for measuring BrC and to assess the magnitude of the various parameters needed for the comparison. Here we focus on the Williams Flats fire measurements on 7 Aug. 2019 as a typical example of the data collected near fires. Detailed calculations are shown for a single plume transect made between 23:34 – 23:39 UTC, which corresponds to the first sampling transect nearest the fire (see Figure 1).

To make the comparison, Mie theory and the size distribution data measured in this plume transect were used to determine, $K_\lambda$, the conversion factor for estimating the particle light absorption coefficient from the solution data, as described in the Methods section. The results, plotted in Figure 4, shows $K_\lambda$ as a function of wavelength. A sensitivity analysis showing the range in $K_\lambda$ predicted due to variability in the various Mie theory inputs is shown in Figure S3.

The contribution of each component to the predicted overall light absorption as a function of wavelength ($b_{ap,predicted,\lambda}$, Eqn (4)) is shown in Figure 5a and the total compared to the PAS data. Similar plots to Figure 5a for each plume transect of the 7 Aug. 2019 Williams Flats fire are given in the Supplemental Material Figure S4. In Figure 5a, the dotted black line is the "bare" (pure) rBC absorption determined from the rBC mass concentration measured by the SP2 (rBC concentration was 4.8 $\mu g\ m^{-3}$) determined by Eqn (3). The effect of the BC coating enhancement ($E_\lambda$=1.6) is also shown, and the resulting overall



BC light absorption is the solid black line. Compared to the PAS data, this coated rBC absorption contributed 61% to the total light absorption at 664 nm, but only about 36% at 405 nm due to BrC contributions. The brown line in Figure 5a is $b_{ap,predicted,\lambda}$, or the ambient light absorption coefficient of TS BrC measured by the LWCC after applying $K_\lambda$ (conversion of solution to particle BrC, $b_{ap,TSBrC,\lambda} = K_\lambda \cdot Abs_{TS,\lambda}^{LWCC}$) added to the BC absorption (first term in Eqn (4)). The brown shading is then the aerosol particle TS BrC ($b_{ap,TSBrC,\lambda}$). The red diamond markers in Figure 5a are absorption measurements made by

the PAS at three wavelengths, representing the overall aerosol absorption. The red curve is from fitting the three PAS measurement points with a power law ($b_{ap,PASBrC} \sim \lambda^{-3.02}$). Note that the area between the red curve and the solid black curve is the estimated PAS BrC absorption from Eqn (2).

In Figure 5b and 5c, a direct comparison is made between PAS BrC ($b_{ap,PASBrC,\lambda}$) and TS BrC ($b_{ap,TSBrC,\lambda}$) as a function of

25 wavelength. At a wavelength near 400 nm the two methods give nearly the same absorption coefficient, but at lower wavelengths, the BrC predicted from the solvent extract is increasingly higher than the PAS-predicted BrC. At wavelengths higher than 400 nm, the PAS BrC tends to be higher by a relatively consistent value. If no conversion from solution to particle absorption factor is applied, (i.e., $K_\lambda$=1), the PAS BrC is consistently higher for all wavelengths greater than about 325 nm, see Figure S5. We note that the predicted overall light absorption is sensitive to the various parameters used in the calculation,

and the agreement at 400 nm is a function of our selected variables.

Although Figure 5 shows data from just one transect through the plume from a single fire (see Figure S4 for all transects in the first lawn mowing pattern flown in Figure 1), these differences between TS BrC and PAS-predicted BrC are consistent between many of the fire plumes investigated in this study. Figure 6 show scatter plots comparing the soluble BrC ($b_{ap,TSBrC,\lambda}$)

and the PAS BrC ($b_{ap,PASBrC,\lambda}$) at the PAS measurement wavelengths (405 nm, 532 nm, 664 nm) for all smoke plumes. From the linear regression, there is a good correlation between the two methods for determining BrC absorption coefficients, with the highest correlation for the lower wavelength (405 nm), where BrC absorption is a larger fraction of the overall light absorption and BrC absorption coefficients are highest (Figure 6a). The data are more scattered and the slope larger (i.e., greater discrepancy) as the wavelength increases from 405 nm to 664 nm.

The wavelength-dependent differences in the soluble BrC and that estimated from the PAS data, shown in Figure 5b and 5c, may be due to a number of factors:

1. Measurement artifacts and uncertainties or differences in the particles size ranges measured by the various instruments. Artifacts related to volatility of BrC for the filter measurements are possible, but we found no large

bias between online and filter measurements of water-soluble BrC (Zeng et al., 2020). Many of the other uncertainties are likely a source of variability in the comparisons, but not a cause for the systematic trends.



2. Uncertainty from the conversion factor $K_\lambda$, which is sensitive to the BrC-containing-particle size distribution and the real part of the refractive index ($n$) (see Figure S3). The conversion factor is most sensitive to $n$. We used a constant value of $n=1.55$, but higher values have been recorded recently in fresh smoke ($n=1.64$ at 475 nm and 1.61 at 365 nm) (Womack et al., 2021). Higher $n$ increases $K_\lambda$ nearly proportionally (Figure S3).

3. Uncertainty in the BC absorption enhancement $E_\lambda$, which is associated with aerosol morphology, including aerosol geometry, shell thickness, and shell optical properties. $E_\lambda$ may vary with wavelength, which we did not consider.

4. Contributions of non-soluble species, such as those expected to be characterized as S-BrC. Larger molecular weight chromophores often absorb more into higher wavelengths and are likely less soluble. Thus, missing non-soluble species in the extracts, but which are included in the PAS BrC, would lead to increasing under-measurement of TS BrC at the higher wavelengths and likely add variability, as observed (Figure 6c). However, there is no correlation between the difference in soluble vs. PAS BrC at the higher wavelength (664 nm) as a function of MCE. Note that in Figure 6c, the ratio between the two BrC measurements can be very high (all data are far above 1:1 line), which is hard to explain by measurement uncertainties or variations of $K_\lambda$ and $E_\lambda$.

5. Narrower wavelength range of the PAS. The PAS data at three wavelengths may not be sufficient to accurately extrapolate absorption beyond the measurement range, especially to lower wavelengths where BrC aerosol predominantly absorbs light. In this case, the PAS data may also not be well characterized by a simple power law fit ($b_{ap,PAS,\lambda} \sim \lambda^{-AAE}$). Jordan et al. (2021) suggest fitting with a second-order polynomial function for $\log(b_{ap,PAS,\lambda})$ vs $\log(\lambda)$, but this does not change the discrepancy at low wavelengths, (blue curve in Figure 7). Adding a data point at a lower wavelength, the predicted BrC at 300 nm ($b_{ap,predicted,300nm}$), to the PAS data and then fitting with a power law tends to produce better agreement with the overall predicted light absorption, although there are still some discrepancies, especially at mid-visible (500 nm) wavelengths. These results suggest that particle absorption instruments that do not measure below wavelengths of ~ 400 nm may significantly under-predict particle absorption contributions when the data is extrapolated to lower wavelengths, if significant levels of BrC are present.

In the following, we use soluble BrC at 365 nm ($Abs^{LWCC}_{365nm}$) (although the conversion factor K is not applied for simplicity) and PAS at 405 nm to investigate causes for BrC variability in plumes, justified by good agreement between the two methods at ~400 nm (Figure 6a). Both these measurements of light absorption are normalized by ΔCO to determine the corresponding NEMR.



### 3.4 BrC evolution

#### 3.4.1 Overall trends in BrC

Starting from a wide perspective we compare the down-wind evolution of TS BrC ($Abs_{TSBrC,365nm}^{LWCC}$) of the FIREX-AQ data to the larger scale evolution of smoke (RIM Fire) reported by Forrister et al. (2015) from a previous study. The same filter-based measurement and analytical methods were used in both cases to determine TS BrC. Analysis of the uniquely large RIM fire that was studied on two separate days as it advected from California into Manitoba, Canada, showed a consistent decrease in the NEMR$_{TSBrC}$ (Figure 8), and an observed half lifetime of TS BrC of 9 to 15 hours was estimated. We have added to the RIM fire data the various measurements from this study.

First noting the more aged (10 to 30 hours) FIREX smoke data in Figure 8, which were identified as smoke from the Tucker and two Williams Flats fires. Some of this data (Tucker and some of 8/07 Williams Flats) tend to follow the steady decay of the RIM fire, but other measurements (all of 8/08 and some of 8/07 Williams Flats) have significantly higher NEMR$_{TSBrC}$ at ages between 15 and 30 hours. For these data, a trend is less clear. Since these fires were not tracked continuously, there is a significant measurement gap of over 10 hours in the plume evolution, the aging process for these smoke plumes is uncertain so causes for the NEMR$_{TSBrC}$ variability cannot be assessed. However, in general, NEMR$_{TSBrC}$ for aged plumes was lower than fresh plumes, suggesting an overall decay of BrC on time scales greater than about 8 hrs.

Focusing on the higher density of measurements made closer to the fires where the transport ages were less than approximately 8-hours, Figure 8 shows that the RIM BrC data are within the range of the FIREX data, but the FIREX NEMR$_{TSBrC}$ data are highly variable with no clear trend with increasing plume age. Looking at each of the fresh plumes investigated in this study, Figure 9 shows that a range of behaviors is seen. No consistent pattern of production nor depletion of BrC is observed in the FIREX-AQ data. In some plumes the data are highly scattered, in others there appears to be a consistent downward or upward trend, or no change in the NEMR$_{TSBrC}$ with increasing time. These trends are similar when using PAS BrC data averaged to the filter sampling times. A similar lack of consistent trends in WS BrC was observed from the Twin Otter measurements as part of FIREX-AQ (Washenfelder et al., 2022, submitted) and the WE-CAN study (Amy Sullivan, personal communication). Waxing and waning of the fire emissions, which were confirmed with the geostationary satellite fire radiative power (FRP) measurements during the period of the DC-8 sampling, and changes in source emission strength (and perhaps aerosol composition) during this time may also impact the downwind variability of the smoke plume (in addition to the dilution, photochemistry, and semi-volatile partitioning processes) (Wiggins et al., 2020; Hodshire et al., 2019). The limitations of semi-lagrangian sampling adds complexity, but the results suggest highly complex and variable BrC evolution. To investigate the changes in NEMR$_{TSBrC}$ we look at the evolution of a specific BrC chromophore and study the variability of species along cross-plume transects.





### 3.4.2 Evolution of bulk BrC compared to 4-Nitrocatechol

One approach to evaluate the evolution of bulk BrC is to compare NEMR$_{BrC}$ as a function of plume age to a specific BrC species with known properties. 4-Nitrocatechol (4-NC) has been observed to be abundant in a variety of BrC sources, including primary emission from biomass burning (Lin et al., 2016) and in the secondary aerosols generated from aromatic precursors (Lin et al., 2015b; Vidović et al., 2020), and its evolution has been studied in detail (Zhao et al., 2015). For FIREX, 4-NC mass concentration was measured on 17 of the 23 flights. As it is one of the components of WS BrC, we compare 4-NC to WS BrC measured in the filter extract. We also compare it to PAS BrC in the aerosol particle phase; both are shown in Figure 10. For all available data within smoke plumes, the ratio of the 4-NC absorption coefficient to the WS BrC absorption, both at the wavelength of 365 nm, was 18% ± 16% (Mean ± Stdev), with lower, middle, and lower quartile range of: 10%, 19%, 33%. Figure 10a shows the statistics of this ratio for data grouped by estimated plume age. In the first 3 hours following emission 4-NC contributed about 23% (median) to the WS BrC light absorption at 365 nm, although there was significant variability. For smoke plumes in the 3 to 6-hour age range the fraction of 4-NC to WS BrC decreased, with a median of about 18%, and for plumes with transport ages greater than 18 hours, 4-NC was essentially all lost, it contributed only about 0.2% to the light absorption of WS BrC. The absorption ratio of 4-NC and all BrC (PAS BrC) at 405 nm wavelength shows a similar decay in the contribution of 4-NC with smoke age. This indicates that the bulk BrC in these smoke plumes had a lifetime that was significantly longer than the 4-NC. As one of the smaller (in terms of molecular weight) BrC chromophores, 4-NC has been found to have a short lifetime of ~4 min from aqueous OH ($3.2 \times 10^{-14}$ M) oxidation after the photo-enhancement stage (Zhao et al., 2015).

### 3.4.3 BrC volatility: A descending plume with increasing temperature

Unlike BC, which is refractory, OA has a wide range of volatility (Huffman et al., 2009). Some chromophores that contribute to the overall BrC may also be semi-volatile and evaporate when the temperature increases, or when the plume dilutes, however, this behavior cannot be inferred from the OA evolution since BrC is only a small mass fraction of OA. For the Sheridan fire on 15 Aug. 2019, a sampling transect was made along the direction in which the plume was advecting away from the fire. In this particular case, the plume descended as it moved away, resulting in a ~15 K temperature increase from the higher to lower altitude, providing an opportunity to investigate the evolution of BrC in terms of temperature-driven evaporation.

Variation in temperature and the NEMRs (to account for dilution) of various species along the plume as it descended are shown in Figure 11. The NEMR$_{rBC}$ (black line) was fairly constant along the plume transect, implying that the combustion conditions (flaming vs smoldering) were relatively stable over time (i.e., since different down-wind distances are related to different times of fire emissions). Thus, we assume that any observed changes with age were mainly from temperature-driven processes as the contribution of chemical aging should be negligible during this time period. Using the higher frequency PAS BrC data, NEMR$_{PASBrC}$ (405 nm) essentially did not change along the plume; the coefficient of variation of NEMR$_{PASBrC}$ was less than





1%. As noted above, 4-NC is less stable than bulk WS BrC, and in this plume it also displayed evidence of volatility-driven loss as advected down-wind, here, at an average rate of 5% $K^{-1}$. For just this data, the 4-NC contributes ~10 % of the total particle BrC absorption closest to the fire and the fraction decreases to ~3% when temperature increases from 287K to 303K. To maintain a constant $NEMR_{BrC}$ despite a substantial decrease of 4-NC, there must be BrC production compensating for the evaporation loss. Perhaps gas phase 4-NC and other volatile chromophores were oxidized to some less volatile species that

partitioned back to the aerosol phase (Roman et al., 2021), or a temperature effect on the particle chemistry, resulting in an unchanged BrC absorption.

### 3.4.4 Possible role of $O_3$

In most cases, the DC-8 flew into plumes approximately perpendicular to the direction of smoke transport, generating

transverse plume transects, as shown in Figure 1. We use these transects with the PAS BrC data to investigate variables that contribute to BrC variability. The filter BrC data could not be used in this analysis since one filter was collected for each transverse transect. Multiple processes, including evaporation and chemical reactions, that can be occurring simultaneously may be easier to resolve in a transverse transect analysis. In an idealized transect of a smoke plume, the aircraft would enter the plume from background air, then experience a positive concentration gradient from edge to plume center, then negative

gradient from center to the other edge, and finally exit the plume into background air. Burning is typically not an ideal point source that produces a plume which fans out as advected away, but often occurs in a region or along a line. Smoke generated at different rates along the whole burning area would then contribute to the concentrations of smoke species measured along the transect. If burning conditions or material burned varied in the region this would complicate the analysis. To minimize this effect on aerosol properties, we focus on the analysis of three contrasting plume transects where in all cases the $NEMR_{rBC}$

was relatively constant (coefficient of variation of $NEMR_{rBC}$ < 10%), suggesting minimal variation in overall particle emissions along the transect. The three plumes investigated are shown in Figure 12.

Figure 12a shows one transect of the Williams Flats plume on 7 Aug. 2019. The CO data suggest smoke from 3 major burning regions had merged into a single plume; three peaks in CO were observed and these plumes had merged since background CO

concentrations were not reached in the regions between the plumes. Figure 12d is one transect of the Castle plume on 12 Aug. 2019, and Figure 12g is one transect of the Williams Flats plume on 3 Aug. 2019. In both of these cases, smoke from two intense burning regions had merged to some extent, based on the CO data. Note the differences in CO concentrations indicating the contrasting levels of emissions from these fires. In most of these cases, the BrC (PAS BrC absorption at 405 nm) profile along the transect had the same shape as CO, suggesting that BrC and CO had the same source, and experienced a similar

dilution process, but there were differences. Figure 12b, 12e, and 12h show $NEMR_{rBC}$ and $NEMR_{PASBrC}$, which removes the effect of plume dilution. $NEMR_{rBC}$ is relatively constant, suggesting that rBC emissions for these fires did not significantly change, (e.g., flaming vs smoldering). For BrC, if the $NEMR_{PASBrC}$ behaved as $NEMR_{rBC}$, it would suggest little net effect of any atmospheric processes, other than a simple dilution effect on BrC concentration, or that during the dilution process





production balanced loss, but the $NEMR_{PASBrC}$ did vary to different extents in these three cases, and the variation was correlated
with $O_3$.

In the 7 Aug. Williams Flats (Figure 12a) and Castle Fire transects (Figure 12d), $NEMR_{PASBrC}$ and $O_3$ concentration had a good
positive correlation (Figure 12c and 12f), suggesting $O_3$ oxidation or related process (e.g., secondary processes) could possibly
be linked to the observed BrC enhancement, indicated by the increasing $NEMR_{PASBrC}$. In the Williams Flats 7 Aug. transect,
the $O_3$ concentration was lowest in the center of the plumes (45, 12 and 29 ppbv, respectively for the three CO peaks) and
higher in the regions where the plumes mixed and CO was lower ($O_3 > 60$ ppbv). Lower $O_3$ in the plume centers was likely
due to $O_3$ titration by $NO_x$ with $NO_2$ photolysis too low to regenerate $O_3$, consistent with the anticorrelation between $O_3$ and
$NO_x$ clearly seen in this transect (Figure 12a). Higher $O_3$ production in the diluted edges of the plume is discussed in detail by
Xu et al. (2021), Wang et al. (2021), and Decker et al. (2021). $NEMR_{PASBrC}$ also tended to be higher in the edge regions between
the plumes where $O_3$ was higher leading to a positive correlation between $NEMR_{PASBrC}$ and $O_3$ (Figure 12c). Concentrations
of various species in this plume (08/07 Williams Flats, Figure 12a, 12b, 12c) were much higher than the other two fires shown
in Figure 12.

For the Castle fire transect (Figure 12d, 12e, 12f), smoke levels were much lower (much lower CO), $O_3$ may not have been
significantly titrated by $NO_x$ (note low $NO_x$ levels). $O_3$ was about ~60 ppbv across the plume, but in this case, there was a
slight enhancement in the center of the plume (Figure 12d), along with $NEMR_{PASBrC}$ (Figure 12e), again leading to a positive
correlation with $O_3$ (see Figure 12f). For the Williams Flats fire on 3 Aug. 2019 (Figure 12g, 12h, 12i), which was more intense
than the Castle fire but less than 08/07 Williams Flats (compare CO), $O_3$ was higher in the center of the plumes along with
$NO_x$, but $NEMR_{PASBrC}$ was lower, being higher at the edges, leading to a negative relationship with $NEMR_{PASBrC}$. A positive
relationship between $O_3$ and $NEMR_{PASBrC}$ may be linked to BrC photo-enhancement ($O_3 \uparrow \rightarrow NEMR_{BrC} \uparrow$, or $O_3 \downarrow \rightarrow NEMR_{BrC}$
$\downarrow$) while an inverse relationship is indicative of photo-bleaching of BrC ($O_3 \uparrow \rightarrow NEMR_{BrC} \downarrow$, or $O_3 \downarrow \rightarrow NEMR_{BrC} \uparrow$). These
two possible divergent behaviors when BrC is oxidized by $O_3$ have been observed in other studies. Sareen et al. (2013) observed
this behavior for secondary BrC (formed with methylglyoxal and ammonium) and Fan et al. (2020) for aerosols from biomass
burning. $O_3$ could also just being acting as a tracer for other oxidation processes.

To look for evidence of these trends in all the data, for each plume transect the relationship between $O_3$ and $NEMR_{PASBrC}$ was
determined, and then grouped as either a positive or negative relationship between $NEMR_{PASBrC}$ and $O_3$. The transect-average
$NO_x$ concentration was then compared for these two groups; results are shown in the boxplot in Figure 13. When $NEMR_{PASBrC}$
had a positive relationship with $O_3$, consistent with $O_3$ enhancing BrC absorption by generating additional BrC chromophores
or transforming BrC to more strongly absorbing compounds, high $NO_x$ was more likely to be present. When $NEMR_{PASBrC}$ had
a negative relationship with $O_3$, suggesting $O_3$ contributed to bleaching of BrC, $NO_x$ concentrations were generally lower.
These observations are consistent with some previous studies. Liu et al. (2015b) found that the presence of $NO_x$ was associated





with the production of organonitrogen compounds via $O_3$ oxidation, such as nitro-aromatics and organo-nitrates, which enhanced light absorption. However, other studies show fragmentation of chromophores on exposure to $O_3$ in a $NO_x$-free environment led to a decrease in BrC absorption (Pillar-Little and Guzman, 2017; Sun et al., 2019). Additionally, the reaction of $NO_2$ with $O_3$ produces the $NO_3$ radical, which has been identified to be an important factor in BrC formation at night (Cheng et al., 2020; Mayorga et al., 2021; Selimovic et al., 2020). In dark optically thick smoke plumes where the nitrate radical loss by photolysis may be suppressed, high $O_3$ and $NO_x$ could also be linked to increases in BrC (Cheng et al., 2020; Mayorga et al., 2021; Selimovic et al., 2020). The correlation is not perfect, since as seen in Figure 13, (left box-whisker plot), there were many periods when BrC increased with increasing $O_3$, and yet $NO_x$ levels were very low. This may be a limitation with our analysis, but there is some evidence that BrC can be formed without $NO_x$ through heterogeneous reactions of ozone with combustion particles (i.e., soot) (Kuang and Shang, 2020). Overall, the range of possible results demonstrate the complexity of processes that may affect BrC in fairly fresh wildfire smoke.

Dilution-driven evaporation resulting in BrC loss has been reported to be an important process in the WE-CAN airborne study, which investigated similar western US wildfires in the summer before FIREX-AQ (Palm et al., 2020). Our analysis comparing the evolution of WS BrC to 4-NC and the change in BrC with changing plume temperature, however, suggested that the dominant fraction of BrC was not volatile. Also, if dilution had a large affect in the 3 plumes above, it is likely that it would have been difficult to discern any trends between NEMR$_{PASBrC}$, $O_3$, and $NO_x$, which implies that dilution played a minor role compared to the effects of $O_3$ on BrC. For example, along these transverse transects, air masses experienced different degrees of dilution; air masses at the edge of the plume, or where two plumes had merged, are more diluted with background air than those at the center of the plumes. In the transect from the Williams Flats fire on 7 Aug. 2019 (Figure 12a), the highest CO mixing ratio was ~5600 ppbv, and the lowest was ~1500 ppbv near where two plumes had intersected, but still sampling in smoke (i.e., CO still significantly above background levels). This corresponds to a dilution ratio (the ratio of highest CO enhancement to the CO enhancement at a location of interest) of about 4. If only considering the effect of dilution-driven evaporation, the NEMR$_{PASBrC}$ profile would be similar to the CO profile (the CO change indicates degree of dilution between two regions). But the profile of NEMR$_{PASBrC}$ was opposite of this, which means other processes, possibly $O_3$ oxidation in this case, drove the change in BrC absorption. This opposite pattern between CO and NEMR$_{PASBrC}$ also occurred in the transects of the Williams Flats fire on 3 Aug. 2019 (Figure 12g, 12h, 12i), but in this case a possible reason for the observed NEMR$_{BrC}$ shape was due to bleaching, or oxidation of BrC by $O_3$. In the Castle fire transect (Figure 12d, 12e, 12f), the dilution effect was superimposed on the enhancement by $O_3$ oxidation. From the analysis above, we conclude that the effect of $O_3$ oxidation, or a process linked to $O_3$ production, was stronger than dilution.

### 3.4.5. Search for other factors causing BrC changes with plume age

As noted in the Introduction, there are a host of factors that can affect BrC levels in an evolving smoke plume. We examined the FIREX-AQ dataset for other potential factors that might alter the optical properties of BrC, including relative humidity



(RH) and aerosol liquid water content (LWC) for evidence of heterogeneous reactions, OH exposure (product of OH concentration and time), $NH_4$ associated with Ammonium- or Amine-containing BrC production, optical thickness of the plumes ($j_{NO2}$ values), and type of material burning, but no evidence was found for a consistent relationship with BrC evolution. Direct photolysis may also change the optical properties of BrC, however, the wildfire flight transects were made in the late afternoon or in the evening. Additionally, $j_{NO2}$ in the center of the plumes was typically less than 5% of the $j_{NO2}$ level outside of the plume, so direct photolysis may not be a significant factor causing BrC bleaching within these plumes. Late afternoon measurements and dense optically thick smoke plumes could also depress OH oxidation, except in the upper levels and sides of the plume where photochemical OH production would be more likely (Wang et al., 2021). Wildfire smoke generated at different times of day may evolve differently due to the type of oxidants involved and extent of photochemical bleaching in the first few hours (i.e., emissions late in the day or at night versus emissions in the morning or early afternoon) The DC-8 rarely continuously flew at the top or edges of plumes, limiting investigating the effect of OH on BrC aerosol in these more dilute regions. It is also possible that multiple simultaneous processes limited our ability to resolve individual ones. A positive matrix factorization (PMF) analysis did not show any consistent factors, which could be either due to lack of clear processes or that many were highly non-linear and not captured by the PMF analysis.

## 4 Summary

Different methods were used to determine particle BrC as a part of the NASA/NOAA FIREX-AQ campaign targeting wildfires burning in the western US in the summer of 2019. Two methods were focused on in this work: BrC based on absorption of aerosol particle chromophores in liquid solvent extracts from particles collected onto filters and BrC inferred from online measurements of total light absorption by particles in their native state with a PAS. The emission ratio of BrC at 405 nm relative to CO is estimated to be 0.131 $Mm^{-1}$ $ppbv^{-1}$ at 405 nm, and 0.071 $Mm^{-1}$ $ppbv^{-1}$ for water soluble BrC and 0.163 $Mm^{-1}$ $ppbv^{-1}$ for total soluble BrC, both at 365 nm and for light absorption in the extract solution (to convert to aerosol absorption multiply by ~1.75). The unique data set and high levels of BrC in these smoke plumes allowed detailed comparison between solvent and PAS BrC measurements. There is considerable uncertainty in the comparison since it requires estimating the contribution of coated refractory black carbon (rBC) as a function of wavelength to the total PAS-measured absorption and a conversion factor to estimate aerosol particle BrC from measurements of BrC in a solvent extract. For the parameters we used to determine these factors, we found that at about ~400 nm, the two methods provide similar estimates of BrC absorption. However, soluble BrC was consistently higher than the PAS BrC, with the difference increasing with decreasing wavelength from 400 to 300 nm, suggesting extrapolating the PAS-inferred BrC to below the lowest measurement wavelength of 405 nm may significantly under estimate BrC light absorption. In contrast, at wavelengths higher than roughly 400 nm, the PAS-inferred BrC was higher than the soluble BrC, but the difference was highly variable. This difference may be due to chromophores that were insoluble in the solvents utilized (water and methanol) and these insoluble chromophores absorb light more strongly at higher wavelengths than soluble species. These types of BrC species may have properties closer to BC, and




are referred to as S-BrC, (strongly absorbing BrC), by Saleh (2020). Overall, the BrC aerosol in smoke observed during FIREX-AQ are in the class of M-BrC, (moderately absorbing), consistent with most emissions encountered in the campaign having a MCE of around 0.9, the boundary between smoldering and flaming burning conditions.

The evolution of BrC in the smoke plumes was also investigated. No consistent pattern of BrC evolution in the first eight hours following emission was observed. Enhancement, depletion and nearly constant $NEMR_{BrC}$ (Normalized Excess Mixing Ratio of BrC, $\Delta BrC/\Delta CO$), were observed in the various plumes. 4-nitrocatechol (4-NC, a known BrC chromophore) was highly depleted in more aged plumes relative to bulk BrC; after roughly 8 hours most 4-NC was lost. Temperature-driven evaporation (T increase of 15 K) resulted in depletion of 4-NC, but had little effect on bulk BrC. We conclude that the majority of BrC was much more stable than 4-NC. Evidence was found that oxidation by $O_3$ in the presence of $NO_x$ might be an important pathway for BrC enhancement, while BrC was more likely to be bleached by $O_3$ when $NO_x$ levels were low. No other factor was found to be consistently related to $NEMR_{BrC}$.

Although the evolution of smoke in the first few hours following emission is highly complex, a few studies show that over larger time scales there tends to be a consistent loss of BrC, and there was some evidence for this in these plumes, but BrC in many cases was not lost as rapidly as reported by Forrister et al. (2015) for the RIM fire. Additional work focusing on the optical impacts of these aged species is needed, given they can impact radiative forcing on global scales (Zeng et al., 2020) over periods of days to weeks. Similar arguments may apply to smoke toxicity, where human exposures can be dominated by highly aged smoke transported far from the fires (O'dell et al., 2021). The toxicity of very aged smoke may have substantially changed since emissions.

**Data Availability:** FIREX-AQ data can be downloaded from the NOAA/NASA FIREX-AQ data archive: FIREX-AQ DOI: 10.5067/SUBORBITAL/FIREXAQ2019/DATA001.

**Author Contribution:** LZ and RJW designed the project and wrote the paper. LZ, RJW, JMK, JPS, AEP, JP, TR, GSD, JPD, JBN, DP, HG, PCJ, JLJ, ES, JD, LX, RHM, EBW collected and analyzed data. All authors reviewed and provided comments for the paper.

**Acknowledgements:** We thank all pilots and crew of the NASA DC-8 for their role in obtaining the data. We thank Nicholas L. Wagner for PAS aerosol absorption measurement data, Christopher D. Holmes for the plume ages, Hannah Halliday for the MCE data, Amber Soja for the burn fuel type data, and Samuel Hall and Kirk Ullmann for photolysis rate data.

**Financial Support:** LZ was supported by NASA grant no. 80NSSC18K0662 and 80NSSC17K043, RJW by NASA grant no. 80NSSC18K0662. ES and JD were supported by NASA grant 80NSSC18K0631. DP, HG, PCJ, JLJ were supported by NASA



grant 80NSSC18K0630, 80NSSC19K0124 and 80NSSC21K1451. LX were supported by NASA grants 80NSSC18K0660 and
80NSSC21K1704.

'15



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

## Tables

**Table 1.** Summary of processes that produce (enhancement) or remove (bleaching) of BrC following emission.

| Mechanism | Fate | Reference |
|---|---|---|
| Reaction with OH, $O_3$ or direct photolysis | Bleaching/enhancement typically first enhancement and then bleaching | Zhong and Jang (2014); Zhao et al. (2015); Wong et al. (2017); Browne et al. (2019); Fan et al. (2020); Fleming et al. (2020); Harrison et al. (2020); Schnitzler et al. (2020) |
| Aqueous reaction involving or forming carbonyl compounds | Enhancement | Nguyen et al. (2013); Powelson et al. (2014); Kasthuriarachchi et al. (2020) |
| High $NO_x$, $NO_3$, associating with night chemistry | Enhancement | Lin et al. (2017); Jiang et al. (2019); Cheng et al. (2020); Li et al. (2020); He et al. (2021); Mayorga et al. (2021) |
| Dilution-driven evaporation | Bleaching | Palm et al. (2020) |





**Table 2:** Details of the wildfire plumes encountered in the western United States during FIREX-AQ 2019 by the NASA DC-8 aircraft. The Date (month/day) and Time are at the point when the aircraft starting sampling in the plumes, (note that UTC may exceed 24:00 to ensure continuity). Except for plumes in CA and WA, which are in the PDT time zone, local time for plumes encountered in other states is in MDT.

| Date (2019) | Plume Name | Time (UTC) | Local Time | State | Fire Location | Fuels (inciweb) |
|---|---|---|---|---|---|---|
| 7/25 | Shady | 22:45-23:26 | 16:45-17:26 | ID | 43.56, -112.89 | Timber and Tall grass |
|  |  | 23:47-25:08 | 17:47-19:08 |  |  |  |
|  |  | 25:46-26:45 | 19:46-20:45 |  |  |  |
| 7/29 | North Hill | 23:22-24:51 | 17:22-18:51 | MT | 46.75, -111.96 | Tall grass, and medium logging slash |
|  | Tucker | 26:38-28:13 | 19:38-21:13 | CA | 41.73, -121.24 | Timber, brush, and tall grass |
| 7/30 | Tucker (Aged) | 21:30-22:37 | 14:30-15:37 | CA | 41.73, -121.24 | Timber, brush, and tall grass |
| 7/30 | Lefthand | 25:34-27:37 | 18:34-20:37 | WA | 46.93, -120.99 | Logging slash and timber |
| 8/2 | Lick Creek | 25:06-26:10 | 18:06-19:10 | ID | 47.16, -115.91 | Logging slash and timber |
| 8/3 | Williams Flats | 22:20-24:03 | 15:20-17:03 | WA | 47.94, -118.62 | Dead trees, grass, sage, and bitter brush |
|  |  | 24:38-26:14 | 17:38-19:14 |  |  |  |
| 8/6 | Williams Flats | 20:58-21:50 | 13:58-14:50 | WA | 47.94, -118.62 | Timber, brush and short grass. |
|  | Horsefly | 22:37-24:32 | 16:37-18:32 | MT | 46.96, -112.44 | Timber (litter and understory) and medium logging slash |
| 8/7 | Williams Flats (Aged) | 22:05-23:02 | 15:05-16:02 | WA | 47.94, -118.62 | Timber, brush and short grass. |
| 8/7 | Williams Flats | 23:34-24:46 | 16:34-17:46 | WA | 47.94, -118.62 | Timber, brush and short grass. |
|  |  | 25:15-26:12 | 18:15-19:12 |  |  |  |
| 8/8 | Williams Flats (Aged) | 21:51-26:14 | 14:51-19:14 | WA | 47.94, -118.62 | Timber, brush and short grass. |
| 8/12 | Castle | 24:15-26:29 | 18:15-20:29 | AZ | 36.53, -112.23 | Timber (litter and understory) |
|  |  | 27:08-27:53 | 21:08-21:53 |  |  |  |
| 8/13 | Castle | 23:13-24:56 | 17:13-18:56 | AZ | 36.53, -112.23 | Timber (litter and understory) |
|  |  | 25:22-27:10 | 19:22-21:10 |  |  |  |
| 8/15 | Sheridan | 25:06-28:42 | 19:06-22:42 | AZ | 34.68, -112.89 | Grass and Brush |
| 8/16 | Sheridan | 24:48-28:28 | 18:48-22:28 | AZ | 34.68, -112.89 | Grass and Brush |





**Table 3:** Coefficient of determination of linear regression ($R^2$) for periods of sampling in smoke plumes. For the filter data (WS BrC and TS BrC at 365 nm), higher time resolution data are averaged to the filter times, and for all others, the comparisons are for 10 sec merged data. PAS BrC is the inferred absorption coefficient of BrC at 405 nm.

|  | WS BrC | TS BrC | PAS BrC | CO | rBC | OA | HCN |
|---|---|---|---|---|---|---|---|
| TS BrC | 0.83 | | | | | | |
| PAS BrC | 0.83 | 0.92 | | | | | |
| CO | 0.83 | 0.90 | 0.96 | | | | |
| rBC | 0.67 | 0.69 | 0.85 | 0.88 | | | |
| OA | 0.79 | 0.88 | 0.92 | 0.94 | 0.85 | | |
| HCN | 0.67 | 0.67 | 0.83 | 0.88 | 0.88 | 0.76 | |
| CH3CN | 0.71 | 0.66 | 0.83 | 0.88 | 0.90 | 0.76 | 0.96 |





**Figures**

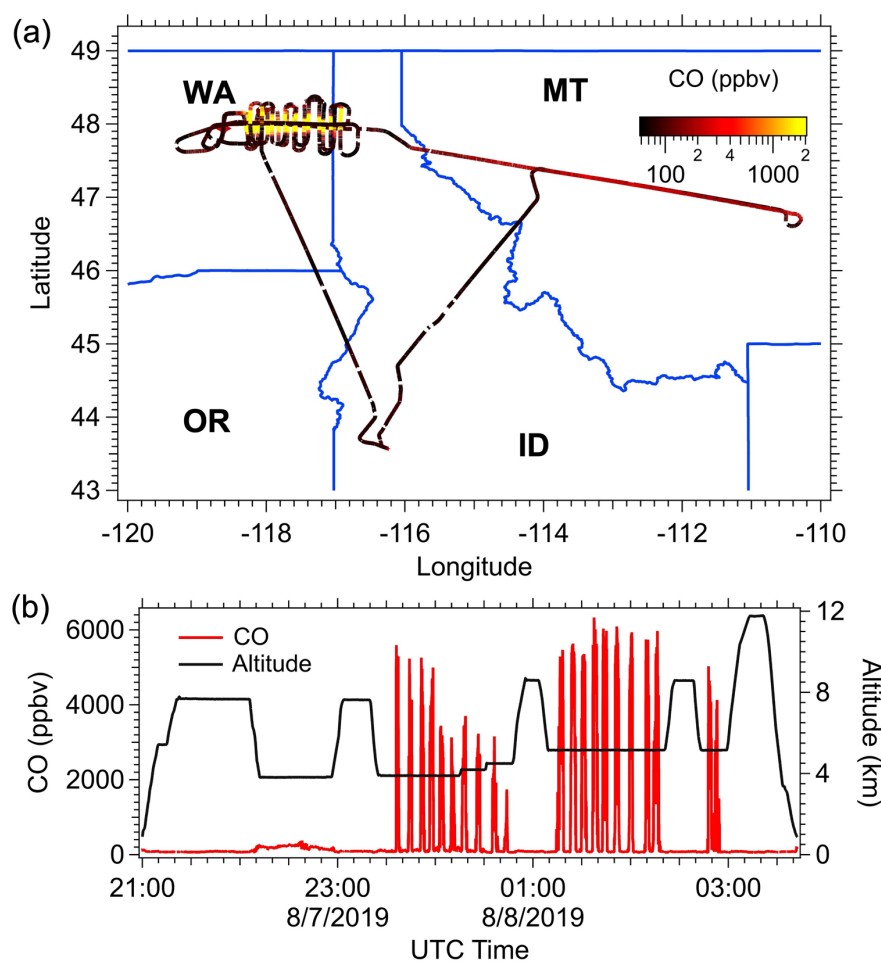

Figure 1 (a) Example flight track on 7 Aug. 2019, targeting smoke plumes emitted from the Williams Flats fire, where the flight track color gives the CO mixing ratio. (b) Time series of CO (red) and aircraft altitude (black).





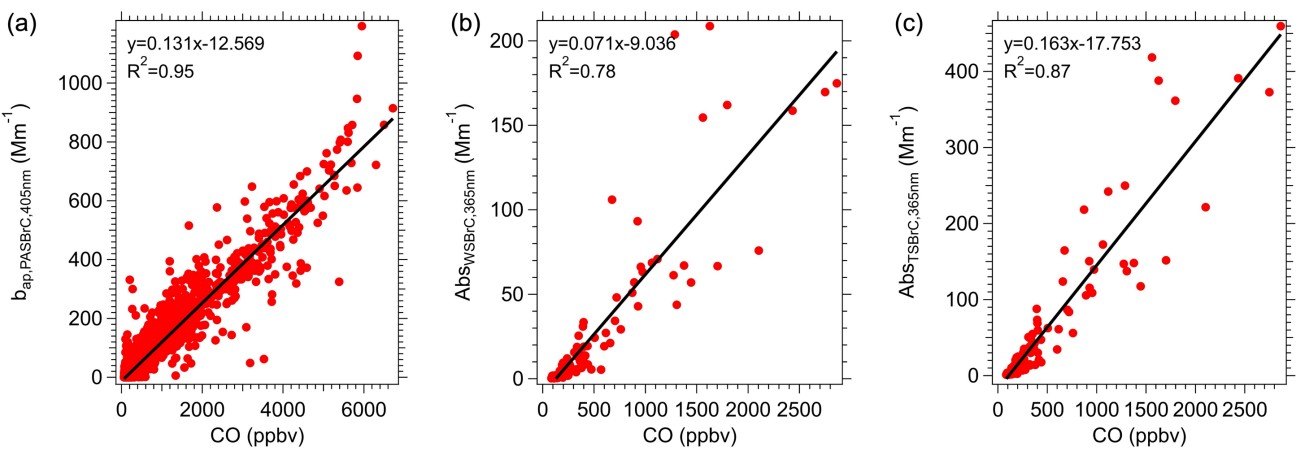

Figure 2. BrC emission ratios (ER) determined from the slope of BrC absorption to CO for the studied fires when the smoke transport time was less than 2 hours. Slopes are from orthogonal distance regression (ODR) of the data. Plot (a) is for PAS data at 405 nm, (b) WS BrC ($Abs_{WS,365nm}^{LWCC}$) and (c) TS BrC ($Abs_{TS,365nm}^{LWCC}$) both at 365 nm. WS BrC and TS BrC ERs are for chromophores in the solvent and have not been converted to aerosol absorption coefficients (see Figure 5 for conversion factor).

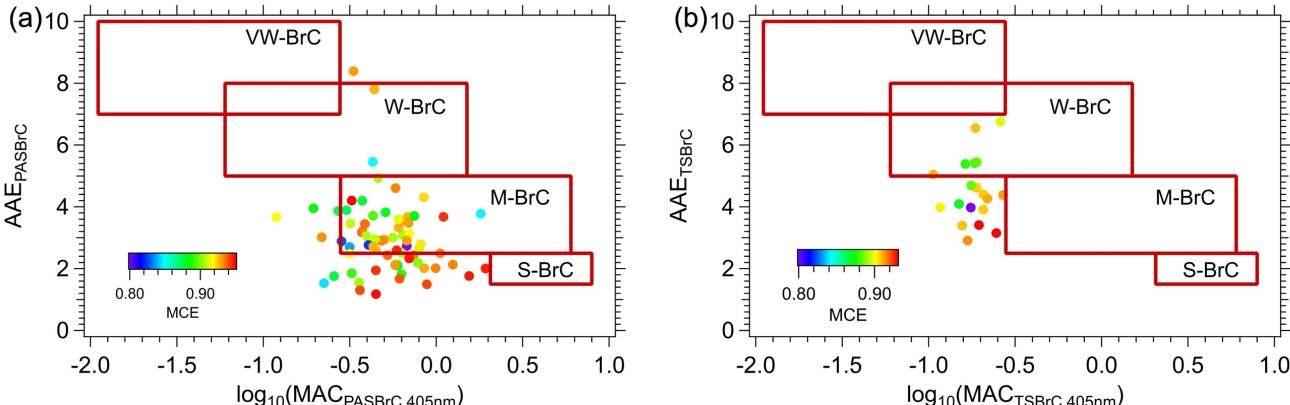

Figure 3. The classification framework proposed Saleh (2020) with wildfire BrC data inferred or measured during FIREX-AQ by (a) PAS and (b) soluble TS BrC. Each datum is one plume transect average. VW-, W-, M- and S-BrC, are very-weakly-, weak-, moderately-, and strongly-absorbing BrC.



05

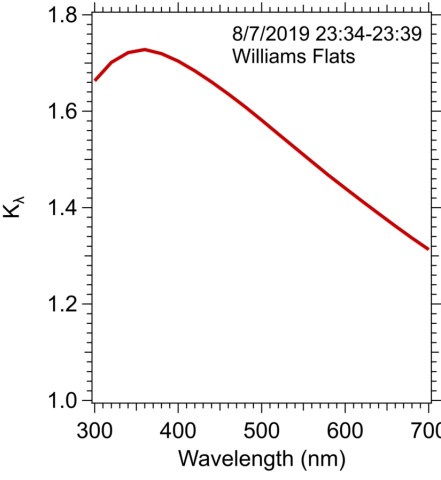

Figure 4. Solution-to-particle light absorption conversion factor $K_\lambda$ versus wavelength calculated from Mie theory for data collected in the first transect of the Williams Flats fire during (23:34-23:39 7 Aug. 2019 UTC).

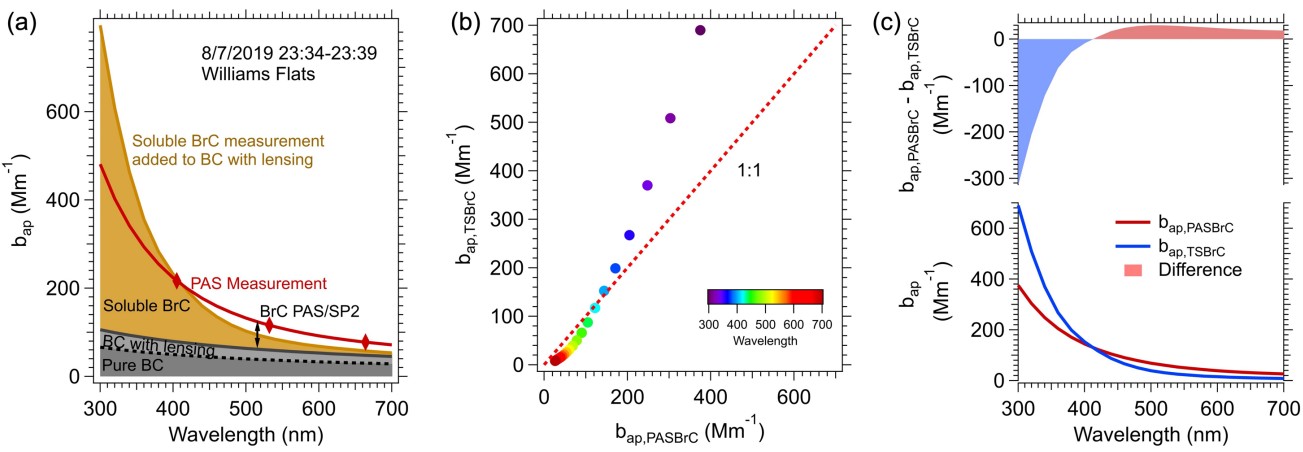

Figure 5. Various light absorption coefficients for the average of the first transect made closest to the Williams Flats fire (23:34–23:39 7 Aug. 2019 UTC). (a) Spectral light absorption closure analysis, where the dashed black line is the light absorption of bare rBC and the solid line is BC considering the coating effect ($E_\lambda$). The brown shading is soluble BrC, $b_{ap,TSBrC,\lambda}$, where $Abs_\lambda^{LWCC}$ was multiplied by the conversion factor $K_\lambda$ to convert from solution to aerosol particle absorption. The upper part of the brown curve is $b_{ap,predicted,\lambda}$, given by Eqn 4. (b) Comparison between $b_{ap,TSBrC,\lambda}$ (brown shading in plot (a)) and $b_{ap,PASBrC,\lambda}$ (difference between red and the black solid line in plot (a)), color coded by wavelength. (c) Similar to plot (b), but versus wavelength (i.e., the difference between BrC determined from the soluble measurements with the conversion factor $K_\lambda$ included, and BrC calculated from the PAS data).





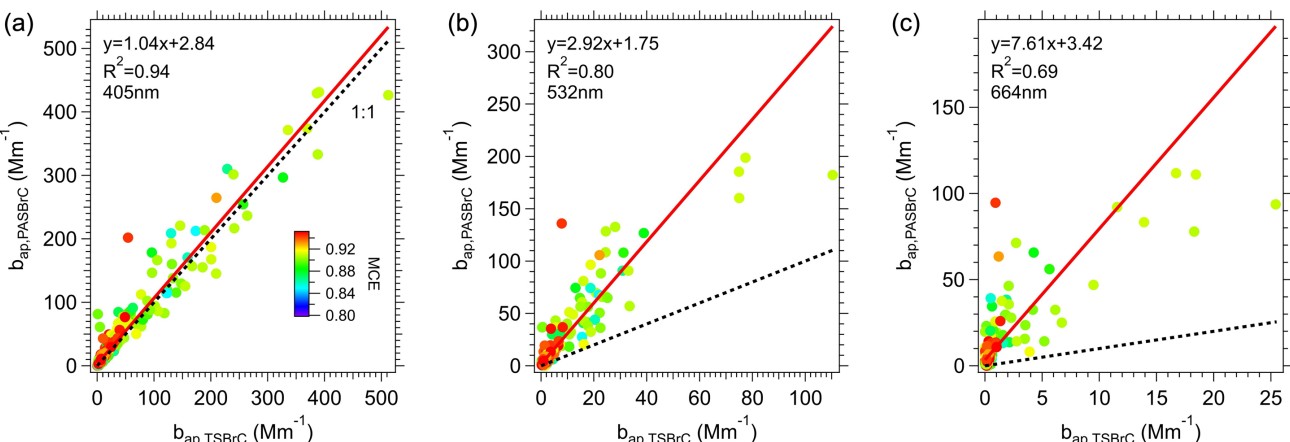

Figure 6. Comparisons between BrC inferred from the PAS ($b_{ap,PASBrC,\lambda}$) and total soluble BrC converted to aerosol absorption ($b_{ap,TSBrC,\lambda}$) at (a) 405 nm, (b) 532 nm, and (c) 664 nm, color coded by MCE. The red line is fitted via orthogonal distance regression (ODR). In all plots, the dotted black line is slope=1.

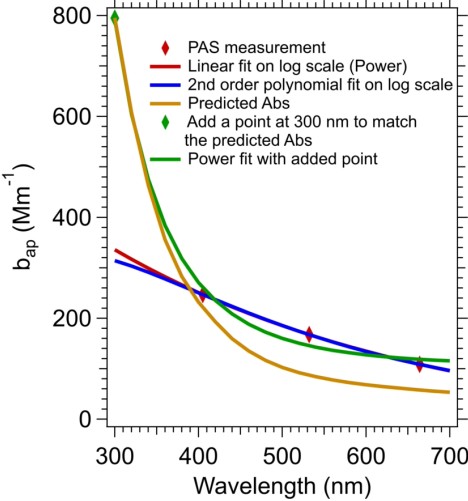

Figure 7. Comparison between the predicted absorption from the sum of BC and TS BrC ($b_{ap,predicted,\lambda}$) brown curve (and in Figure 5a) and various fits to the PAS data (red diamonds). The red line is PAS data fitted with a line on a log-log scale, which is the typical power law fit, the blue curve is a second order polynomial fit on log-log scale, and the green line is a power law with an added data point from $b_{ap,predicted,300nm}$ at 300 nm.





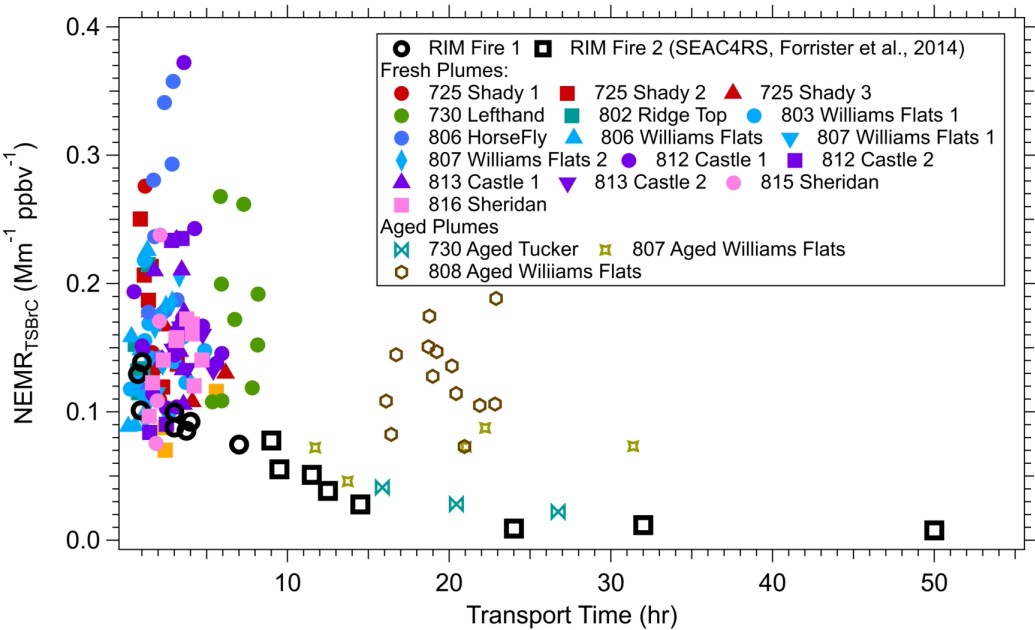

Figure 8. NEMR$_{TSBrC}$ at 365 nm measured in liquid extracts (conversion factor K is not applied) versus smoke transport time. Different colors represent different plumes (also see Table 2). Open markers (circles and squares) are data obtained from the RIM fire during the SEAC⁴RS campaign reported by Forrister et al. (2015).



Figure 9. Evolution of water, methanol and total soluble forms of BrC relative to CO (NEMR$_{BrC}$ = ΔBrC/ΔCO) within each of the various smoke plumes investigated in detail during FIREX-AQ. All of these data on one plot are shown in Figure 8. Each data point is one plume transect (filter sample). Red data points are WS BrC, blue MS BrC, and green TS BrC, which is the sum of the red and the blue. Linear fits are included with the data.





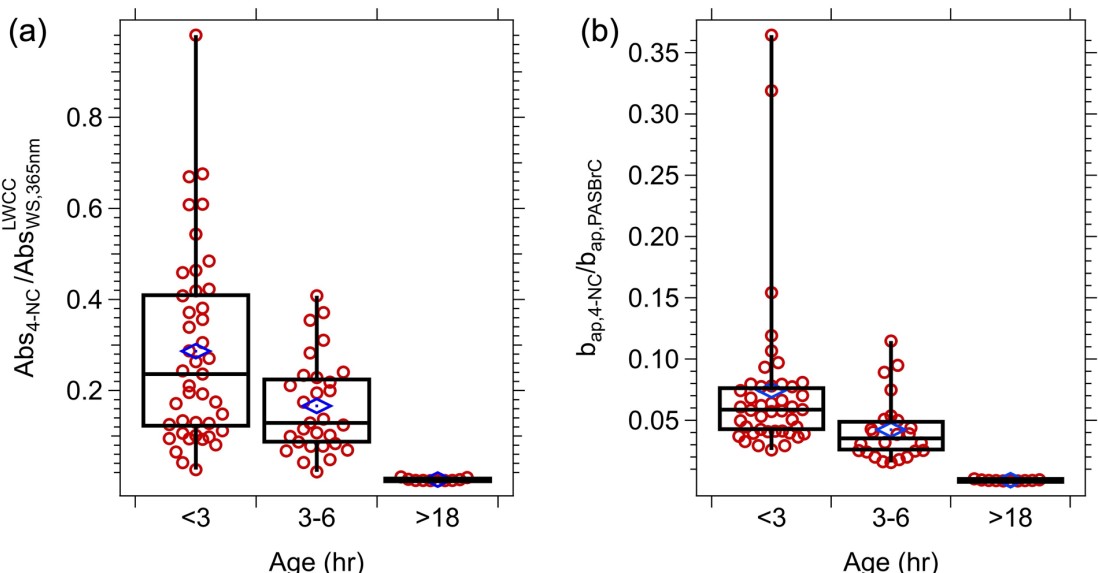

Figure 10. (a) Statistics of the ratio of the absorption coefficient in smoke plumes of 4-NC ($Abs_{4-NC}$) at 365 nm to WS BrC ($Abs_{WS,365nm}^{LWCC}$) for different ranges of transport time. (b) Comparison of the absorption coefficient of 4-NC ($b_{ap,4-NC}$) to BrC ($b_{ap,PASBrC,\lambda}$) inferred from PAS at 405 nm. The MAC of 4-NC used to calculate $Abs_{4-NC}$ and $b_{ap,4-NC}$ is from Zhang et al. (2013), and the conversion factor ($K_\lambda$) from liquid to aerosol ($K_\lambda$) of 1.6 was applied to convert from $Abs_{4-NC}$ to $b_{ap,4-NC}$. Blue markers are means of each bin and in plot (a) are 28.7%, 16.6%, and 0.5% and for plot (b) 7.4%, 4.2%, and 1.1%, respectively.





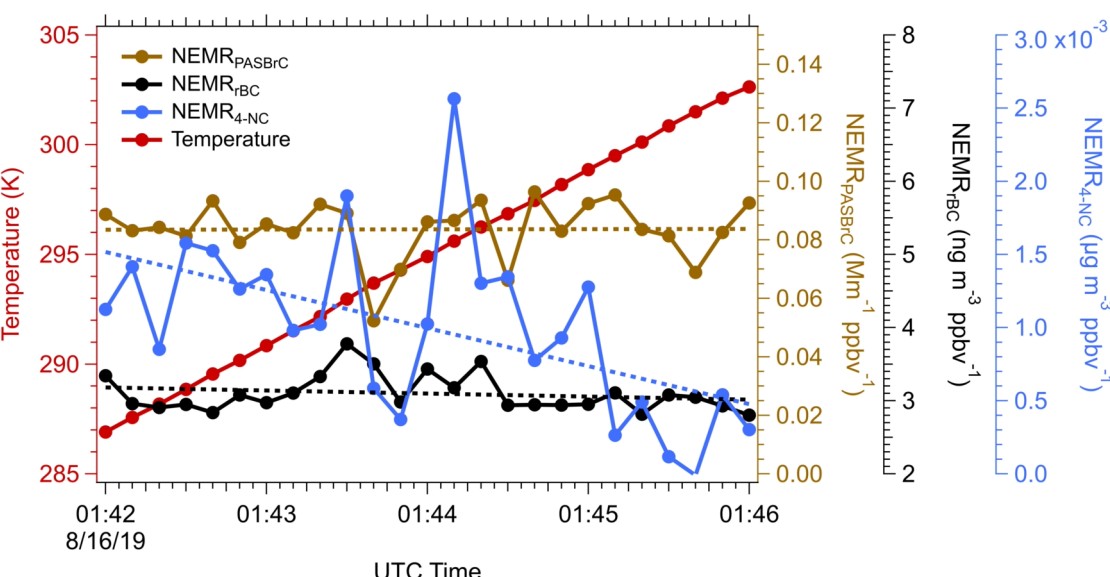

Figure 11. Assessment of volatility of various light absorbing species by comparing the time series of NEMR$_{rBC}$ (black), NEMR$_{OA}$ (green), NEMR$_{PASBrC}$ (brown), NEMR$_{4-NC}$ (blue), for a measurement period when the temperature (red) increased while sampling within a descending smoke plume from the Sheridan fire on 15 Aug. 2019. Dotted lines are trend lines with time fitted by ODR. Data are 10s averages.

65

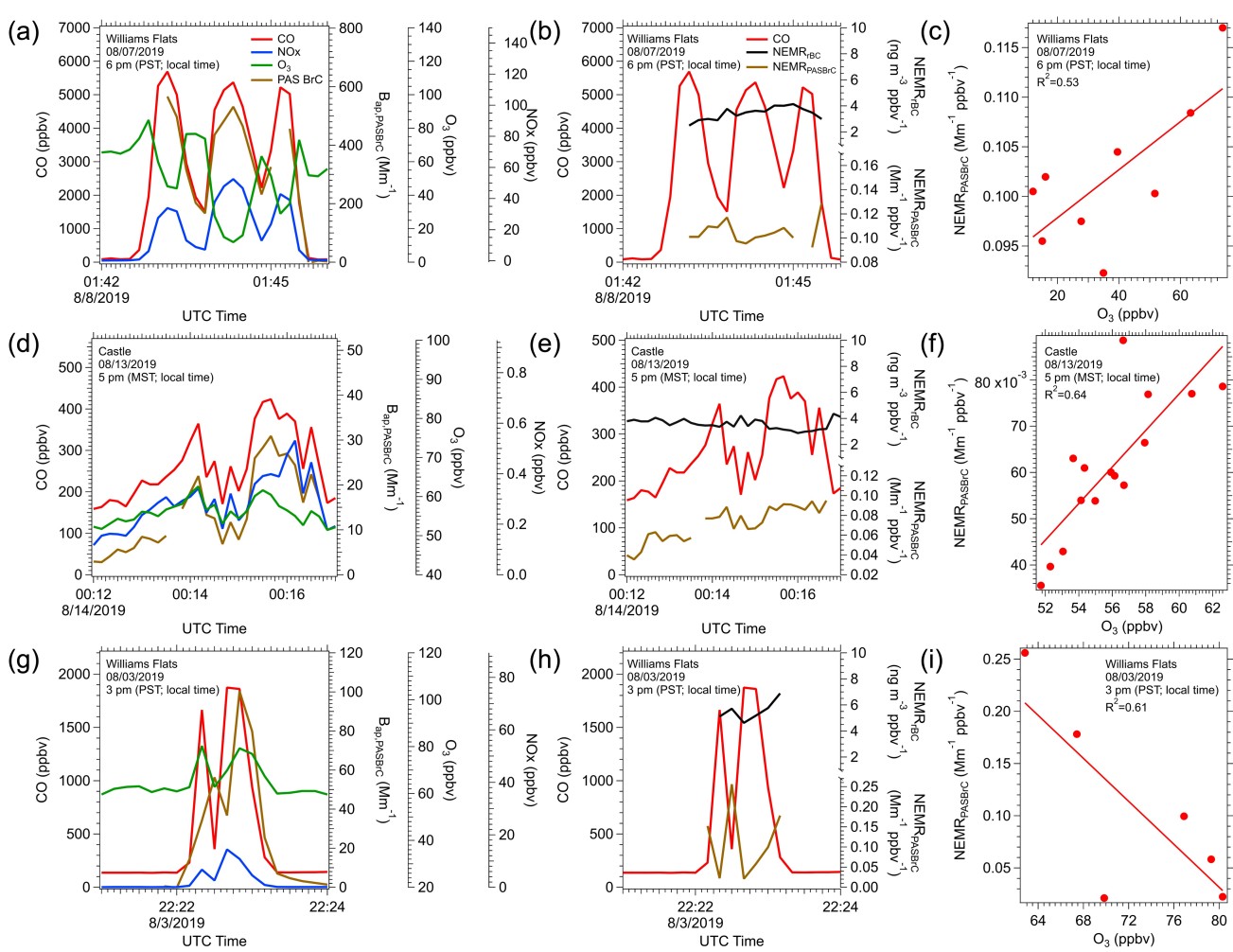

Figure 12. Time series for the concentration of CO (red), BrC from the PAS (brown), $O_3$ (green), and $NO_x$ (blue) in three example plume-transects in plots (a), (d), and (g). Corresponding time series for $NEMR_{BC}$ (black) and $NEMR_{BrC}$ (brown) for these transects, plots (b), (e), and (h), and the relationship between $NEMR_{BrC}$ and $O_3$ for each of these periods of in-plume sampling, plots (c), (f) and (i).



75

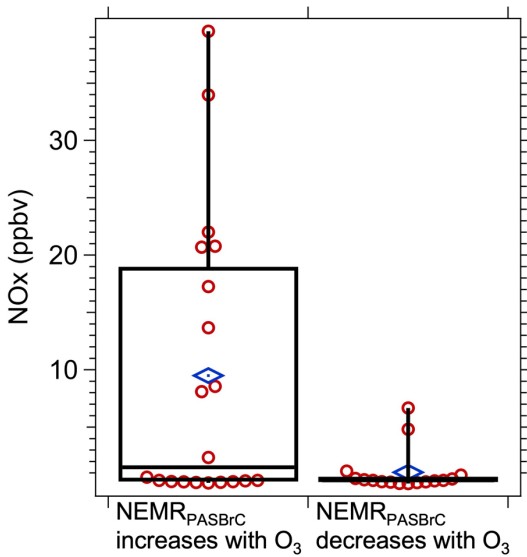

Figure 13. Comparison between the average $NO_x$ level across the transect for two groups of data segregated by the $NEMR_{PASBrC}$ having either a positive or negative relationship with $O_3$, such as shown in Figure 12.

80