# Peer review of "Characteristics and Evolution of Brown Carbon in Western United"

_Atmospheric Chemistry and Physics, 2022_

## Author Response (AR2)

Reviewer 1:

We thank the reviewer for their insightful comments and have responded to each shown in blue text below each comment. Underlined text is the modification to the manuscript.

General comment:

This manuscript presents an investigation of brown carbon (BrC) from wildfire emissions using aircraft plume measurements as part of the FIREX-AQ study. The manuscript has two main foci. The first involves quantifying BrC light-absorption properties and contribution to wildfire aerosol absorption using an online method (which relies on online measurements of absorption coefficients using a PAS and BC concentrations using an SP2) and an offline method (which relies on solvent extraction using both water and methanol). The second involves investigating the evolution of BrC in the atmosphere including (i) overall change in BrC absorption, (ii) comparison of overall change in BrC absorption to 4-Nitrocatechol, (iii) effect of evaporation, and (iv) the role of ozone.

The manuscript is well-written and presents comprehensive high-quality data and analyses, which constitute an important contribution to the understanding of wildfire BrC. I find the following to be particularly interesting: (i) the comparison between offline and online BrC measurements, (ii) the comparison between 4-NC and overall BrC evolution, and (iii) illustrating the complex dynamics that govern BrC evolution. Below is a list of comments that I believe should be addressed in the revised version of the manuscript.

Major specific comments:

1) The comparison between b_ap_PASBrC and b_ap_TSBrC (Figure 6): The difference in comparison at different wavelength is interesting and should be further discussed. The slope of the comparison increases with wavelength, which indicates that (i) there are insoluble BrC species and (ii) these species have a smaller AAE than the TSBrC (which leads to the wavelength-dependent comparison). As the authors point out (Page 14 Line 29), the absolute comparisons (i.e. the slopes in Figure 6) are uncertain. However, the trends are still informative. Starting with the theoretical baseline that b_ap_TSBrC cannot exceed b_ap_PASBrC (because PASBrC represents the total BrC), the slopes in Figure 6 are likely underestimates. Nevertheless, the results highlight the importance of methanol- insoluble BrC, which, according to Figure 6b and 6c, contributes an average of ~65% and ~85% of BrC absorption at 532 nm and 664 nm, respectively. This qualitatively agrees with the findings of (Atwi et al., 2022) that biomass-burning BrC can be split into a less- absorbing methanol-soluble fraction (smaller MAC and larger AAE) and a more-absorbing methanol-insoluble fraction (larger MAC and smaller AAE), with the methanol-insoluble fraction dominating mid-visible absorption.

**Response:** We have added text to increase clarity based on the comments above. It now reads: "These missed insoluble species could come from two sources: 1) Particles containing chromophores insoluble in water, but possible soluble in methanol, that were separated from the particle filter during the first water extraction and then removed by the liquid syringe filter and so not measured, and 2) Particles insoluble in methanol. A likely example of BrC species missed

is tar balls (Corbin et al). Thus, missing non-soluble chromophores in the extracts, but which are included in the PAS BrC, would lead to increasing bias of lower TS BrC at the higher wavelengths and likely add variability, as observed (Figure 6c) as an increasing slope and lower R2 compared to the lower wavelengths. Based on the regression fits in figures 6b and 6c, this implies that at wavelengths of 532 and 664 nm, methanol soluble BrC misses roughly 65% and 87% of the overall light absorption at those respective wavelengths. (Or methanol insoluble BrC chromophores contribute 65 and 87% to the light absorption at 532 and 664 nm, respectively, which are consistent with the findings of Atwi et al. (2022))."

2) Figure 3 and associated discussion: The paragraph on top of Page 13 points out the lack of correlation between MAC and AAE. This is expected because there is no substantial variability in BrC sources and combustion conditions in this study. Due of the usually encountered large spread in BrC data (due to both true variability in optical properties as well as measurement uncertainty), the inverse MAC vs AAE trend becomes apparent only when comparing different BrC categories. For instance, in Figure 1 in Saleh (2020), the inverse MAC vs AAE trend would not be apparent if only looking at one category (e.g. smoldering biomass combustion or SOA from aromatic VOCs).

With that being said, comparing the average MAC and AAE of PASBrC with those of TSBrC would be informative (see (Atwi et al., 2022)). I suggest combining the two panels of Figure 3 in one figure (using different symbols for TSBrC and PASBrC) and showing the average and standard deviation for each group. Doing so will illustrate the inverse MAC vs AAE relation. Specifically, TSBrC will be shown to have a smaller MAC but larger AAE than PASBrC, in agreement with the results of Atwi et al. (2022).

Also, because of the inverse MAC vs AAE relation, the MAC values of different BrC categories start to converge at shorter wavelengths. It is therefore more informative to present Figure 3 at 532 nm, which would better illustrate the difference between TSBrC and PASBrC.

Finally, what is the reason for TSBrC having less data points than PASBrC?

**Response:** Fig. 3 has been replotted with 532 data and BC/OA ratio, as suggested. The large black circles and error bars are mean +/- stdev of the data. We have kept them on two separate plots. The text has been changed to: "PAS data are shown with the Saleh BrC characteristics identified by regions in the boxes. In Figure 3a, these wildfires data are best characterized as M-BrC and shows a weak trend with rBC/OA ratio as higher rBC/OA ratio (more flaming) tends to appear at the bottom left." The sentence in Page 13, Line 90-93 has been deleted. The sentence in Page 13, Line 96 has been changed to "Most of these data are outside of the Saleh's categorization, but they are shifted to the upper left relative to the PAS data, consistent with the idea that PAS BrC contains relatively more weakly absorbing species (Atwi et al., 2022). Like the PAS BrC, TS BrC also does not show a correlation between AAE and $\log_{10}$(MAC) but a weak trend with rBC/OA."

The reason that TSBrC had less data (in the preprint version) was because data were merged into plume patterns (i.e. lawn-mowing patterns). In the new Fig. 3, every filter data collected in the plume is shown. A text has been added to the caption of Fig. 3 to clarify.

[Figure]

Figure 3. The classification framework proposed Saleh (2020) with wildfire BrC data inferred or measured during FIREX-AQ by (a) PAS and (b) soluble TS BrC. Each datum is the average of a plume transect for the PAS data, while all filters collected in the smoke are shown. VW-, W-, M- and S-BrC, are very-weakly-, weak-, moderately-, and strongly-absorbing BrC. The black with error bars is mean ± stdev of the data in each plot.

3) There are several studies that pointed out that MCE does not correlate well with aerosol light-absorption properties (e.g. (McClure et al., 2019; Pokhrel et al., 2016)). I suggest using BC/OA as a proxy for combustion conditions in Figures 3, 5, and 6.

**Response:** Figure 3 and 6 have been re-plotted using rBC/OA ratio. In addition to changes in Comments #2, the sentence in Page 15, Line 57 has been changed to "However, there is no correlation between the difference in soluble vs. PAS BrC at the higher wavelength (664 nm) as a function of rBC/OA". The sentence in Page 22, Line 79 has been changed to "Overall, the BrC aerosol in smoke observed during FIREX-AQ are in the class of M-BrC, (moderately absorbing), and BrC generated from more flaming conditions (higher rBC/OA ratio) tends to be more absorbing but with lower AAE."

[Figure]

Figure 6. Comparisons between BrC inferred from the PAS ($b_{ap,PASBrC,\lambda}$) and total soluble BrC converted to aerosol absorption ($b_{ap,TSBrC,\lambda}$) at (a) 405 nm, (b) 532 nm, and (c) 664 nm, color coded by *rBC/OA ratio*. The red line is fitted via orthogonal distance regression (ODR). In all plots, the dotted black line is slope=1.

4) The manuscript presents details of uncertainty associated with each measurement / analysis, but the uncertainty is not reflected in the figures. Uncertainties can be added as light-gray error bars, which should not have a substantial effect on the clarity of the figures.

**Response:** We decomposed the uncertainty for the optical closure analysis into two parts: 1. from BC, 2. from BrC (Abs$_{TS\ BrC}$*K). The uncertainty from BC was estimated to be 40%, same as the dominant measurement uncertainty of the SP2 instrument as flown in FIREX-AQ. The uncertainty from BrC was derived from TS BrC measurement uncertainty and the uncertainty for the conversion factor K. Error bars representing uncertainties have been added to Fig. 5a. The following caption has been added: "1. The effect or rBC coating, $b_{ap,BC,\lambda}$, dark gray in plot (a), estimated to be 40%, the same as the dominant measurement uncertainty of the SP2 instrument as flown in FIREX-AQ ; 2. TS BrC $b_{ap,TSBrC,\lambda}$, brown, from (Abs$_{TS\ BrC}$*K), considering the uncertainty in measurements and K."

[Figure]

Figure 5. Various light absorption coefficients for the average of the first transect made closest to the Williams Flats fire (23:34–23:39 7 Aug. 2019 UTC). (a) Spectral light absorption closure analysis, where the dashed black line is the light absorption of bare rBC and the solid line is BC considering the coating effect ($E_\lambda$). The brown shading is soluble BrC, $b_{ap,TSBrC,\lambda}$, where $Abs_\lambda^{LWCC}$ was multiplied by the conversion factor $K_\lambda$ to convert from solution to aerosol particle absorption. The upper part of the brown curve is $b_{ap,predicted,\lambda}$, given by Eqn 4. Uncertainties at two extreme wavelengths (300 nm and 700 nm) for two individual components (1. The effect or rBC coating, $b_{ap,BC,\lambda}$, dark gray in plot (a), estimated to be 40%, the same as the SP2 measurement uncertainty; 2. TS BrC $b_{ap,TSBrC,\lambda}$, brown, from (Abs$_{TS BrC}$*K), considering the uncertainty in measurements and K). (b) Comparison between $b_{ap,TSBrC,\lambda}$ (brown shading in plot (a)) and $b_{ap,PASBrC,\lambda}$ (difference between red and the black solid line in plot (a)), color coded by wavelength. (c) Similar to plot (b), but versus wavelength (i.e., the difference between BrC determined from the soluble measurements with the conversion factor $K_\lambda$ included, and BrC calculated from the PAS data).

5) The calculation of PAS BrC absorption coefficients (b_ap_PASBrC) and the corresponding MAC_PASBrC include multiple steps that should be presented in the SI. Please show:

a) rBC size distributions and how they were adjusted to account for rBC outside the detection window of the SP2. Typically, how much rBC was found to be outside of the detection window of the SP2?

**Response:** The following text was added to the manuscript: Accumulation-mode rBC concentrations used for calculation of BC-specific absorption were extrapolated from the SP2 observations to account for rBC mass outside the detection range of that instrument. Log-normal fits to the size distributions of rBC observed by the SP2 were generated on a per-flight basis to allow estimation of the undetected mass in that mode. Corrections were applied evenly to all the rBC concentrations of that flight. Note that the size distributions were strongly dominated by the fire-generated smoke, and thus do not reflect background or urban rBC detected in transit etc. Figure S1 shows a typical rBC size distribution from a flight sampling the Williams Flats fire (August 3, 2019). This generates a corrective scaling factor of 1.12. The average correction factor for all western wildfires in FIREX-AQ was 1.08 with a standard deviation of 0.04 (13 flights).

We have added the Fig below to the Supp.

[Figure]

Figure S1. Mass size distribution of rBC measured for a flight. A log normal fit over the valid range of detection is used to infer the rBC mass undetected in the population.

b) Particle size distributions measured with LAS and the corresponding lognormal fits. How much of the particle number was outside the detection window of LAS? Also, Page 8 Line 48 states that the variability in smoke aerosol refractive index causes an uncertainty of 20%. How is this estimated? In addition to this uncertainty, does the fact that the instrument was calibrated using non-absorbing aerosol (ammonium sulfate) cause any systematic uncertainty because the smoke aerosol is light-absorbing?

**Response:** The size window for LAS is from 100 nm to ~4 um, and an example particle number size distribution is given in Fig. 10 in Moore et al. (2021). Number of particles missed by the LAS (mainly less than 100 nm) were less than 5%. Fig. 1a in Moore et al. (2020) shows the results of the scattering intensity between the Mie theory calculations of sized -resolved particles for ammonium sulfate (m=1.52+0i) and the LAS, which was calibrated with ammonium sulfate, indicate that the LAS tends to undersize particles with real refractive indices less than 1.52, while oversizing particles with larger real refractive indices. Absorbing particles tend to be oversized, as shown in Fig. 1c in Moore et al. (2021). Using clear ammonium sulfate can result in particle oversizing, but this effect is less than 10% since the imaginary refractive index of k is ~0.01 at 365 nm. Therefore, overall uncertainty for the LAS measurements was estimated to be 20% to account for variability in smoke aerosol refractive index. A sentence has been added to Page 8, Line 45 for clarity "The LAS is known to undersize particles with real refractive indices less than ammonium sulfate (m=1.52+0i), while oversizing particles with larger real refractive indices or absorbing particles (Moore et al., 2021). The LAS uncertainty is estimated to be 20% to account for variability in smoke aerosol refractive index and the LAS particle sizing calibrations."

6) Section 3.4.3 and Figure 11: The data collected over this short period of time (4 minutes) does not provide enough evidence to arrive at the conclusion that (i) 4-NC contribution to absorption dropped from 10% to 3% and (i) there must be BrC production (which contradicts the statement earlier in the section that "chemical aging should be negligible during this time period."). For

instance, the variability in 4-NC concentration over a period of 20 s around 1:44 is approximately twice the inferred change over the measurement period (dotted blue line). Also, I would assume that if the difference between NEMR_PASBrC and NEMR_4-NC is plotted on Figure 11, the trend would be very similar to NEMR_PASBrC (i.e. it would not show any increase in non-4-NC BrC absorption).

**Response:** It is difficult to analyze the volatility of BrC aerosol because chemical reactions and physical evaporation occur simultaneously once emitted from fires. According to Zhao et al. (2015), the lifetime of 4-NC under aqueous OH oxidation is ~40 min, and therefore the change of 4-NC would be 10% across this transect, (if the conditions used in their laboratory experiments apply to our field data). Although 4-NC concentration had a larger variation around 1:44 UTC, the regression line between $NEMR_{4-NC}$ and time/temperature was significant, with p value (0.056) less than 0.1. The sentence in the caption of Figure 11 has been changed to "Dotted lines are trend lines with time, fitted by ODR, and p-values for all linear regressions are less than 0.1". The reduction of 4-NC concentration in 4 minutes was more than 50%, which is hard to explain by the reaction with OH. The time series for the absorption of non-4-NC BrC (PAS BrC minus 4-NC BrC) is plotted in the Fig below ; an increase of non-4-NC absorption is hard to discern with R=0.2, p>0.1. Thus, the conclusion about BrC generation has been weakened and the following line added to the end of this section: Firm conclusions are not possible due to the high variability in 4-NC relative to loss trend for this small time period.

[Figure]

Minor specific comments:

1) Line 65: This statement necessitates specifying an imaginary part of the refractive index cutoff above which the OA is said to be light-absorbing. I would rephrase this sentence to reflect the fact that OA is made of components with variable light-absorption properties that vary from negligibly absorbing to strongly absorbing.

**Response:** We have edited the text to now state: "Organic aerosol is made of components with variable light-absorbing properties that vary from negligibly absorbing to strongly absorbing,

with negligibly absorbing being the most common, (i.e., only a small mass fraction of OA appreciably absorbs light)."

2) Line 83: By definition, DRE of a certain component is obtained as the difference in radiative balance with and without the component. The statement here that BrC contributed 7 to 48% of DRE is not consistent with this definition because these values were obtained as the difference in radiative balance with and without BrC absorption, and should more accurately be referred to as 'BrC absorption DRE,' not DRE of BrC (see (Saleh, 2020; Wang et al., 2018)).

**Response:** We have changed the sentence to: "Pole-to-pole BrC measurements through the Atlantic and Pacific Basins showed that for the regions where measurements were made, the top of atmosphere direct radiative effect (DRE) due to BrC absorption ranged from 7 to 48% relative to all light-absorbing carbonaceous particles (BC+BrC), and that most of the BrC was from biomass burning emissions transported over long distances (> 1,000's of km) (Zeng et al., 2020)."

3) Section 3.4.2: Please provide 4-NC optical properties used to calculate absorption coefficients.

Response: A sentence in the caption of Figure 10 has changed to "The MACs of 4-NC used to calculate $Abs_{4-NC}$ and $b_{ap,4-NC}$ are 7.15 and 3.08 $m^2g^{-1}$ at 365 nm and 405 nm, respectively (Zhang et al., 2013), and the conversion factor ($K_\lambda$) from liquid to aerosol of 1.6 was applied to convert from $Abs_{4-NC}$ to $b_{ap,4-NC}$."

References:

Atwi, K., Perrie, C., Cheng, Z., El Hajj, O., & Saleh, R. (2022). A dominant contribution to light absorption by methanol-insoluble brown carbon produced in the combustion of biomass fuels typically consumed in wildland fires in the United States. *Environ. Sci.: Atmos.* https://doi.org/10.1039/D1EA00065A

McClure, C. D., Lim, C. Y., Hagan, D. H., Kroll, J. H., & Cappa, C. D. (2019). Biomass-burning derived particles from a wide variety of fuels: Part 1: Properties of primary particles. *Atmospheric Chemistry and Physics Discussions*, *2019*, 1–37. https://doi.org/10.5194/acp-2019-707

Pokhrel, R. P., Wagner, N. L., Langridge, J. M., Lack, D. a., Jayarathne, T., Stone, E. a., et al. (2016). Parameterization of single-scattering albedo (SSA) and absorption Ångström exponent (AAE) with EC/OC for aerosol emissions from biomass burning. *Atmospheric Chemistry and Physics*, *16*(15), 9549–9561. https://doi.org/10.5194/acp-16-9549-2016

Saleh, R. (2020). From Measurements to Models: Toward Accurate Representation of Brown Carbon in Climate Calculations. *Current Pollution Reports*. https://doi.org/10.1007/s40726-020-00139-3

Wang, X., Heald, C. L., Liu, J., Weber, R. J., Campuzano-Jost, P., Jimenez, J. L., et al. (2018). Exploring the observational constraints on the simulation of brown carbon. *Atmospheric Chemistry and Physics*, *18*(2), 635–653. https://doi.org/10.5194/acp-18-635-2018

Reviewer 2

We thank the reviewer for their insightful comments and have responded to each shown in blue text below each comment. Underlined text is the modification to the manuscript.

This manuscript investigates characteristics and evolution of brown carbon (BrC) using online photoacoustic spectrometer (PAS) that measures dry aerosol absorption of fine particles and offline filter-based approach using liquid spectrophotometric measurements of extracts of particles collected on filters. They compared the measurements at different wavelengths and found that good agreement of BrC absorption at 400 nm. While doing the comparisons, there are several assumptions and limitations, but it still provides useful information and worth publishing. The study claims that investigated samples falls under moderately absorbing class. They also investigated a particular BrC chromophore, 4-nitrocatechol and its evolution with plume ages. Results indicate that 4-nitrocatechol depleted with plume ages, while other BrC was much stable even with increasing temperature in downwind. However, some previous study reported that particulate nitrophenol and nitrocatechol isomers can contribute significantly to BrC absorption at 405 nm in aged wildfire smoke.

This is an interesting study and will be useful for the community. Overall, the manuscript is clearly written, some suggested clarifications are listed below. However, prior to acceptance, the authors should address the following questions/ suggestions and modify the manuscript accordingly.

Specific comments:

The comparison between bap, PASBrc and bap,TSBrC at 405 nm looks good. It might be good to add some discussion why the PAS derived BrC absorption is higher than the TS Brc at higher wavelength. I see that the authors add some discussion about the insoluble chromophores, but it will be good add this discussion in the results section and will be easier for readers to follow.

**Response:** The first reviewer also had a similar suggestion, so we have added some text for clarity. It now reads: *"Thus, missing non-soluble* chromophores in the extracts, but which are included in the PAS BrC, would lead to increasing bias of lower TS BrC at the higher wavelengths and likely add variability, observed (Figure 6c) as an increasing slope and lower $R^2$ compared to the lower wavelengths. Based on the regression fits in figures 6b and 6c, this implies that at wavelengths of 532 and 664 nm, methanol soluble BrC misses roughly 65% and 87% of the overall light absorption at those respective wavelengths. (Or methanol insoluble BrC chromophores contribute 65 and 87% to the light absorption at 532 and 664 nm, respectively, which are consistent with the findings of Atwi et al. (2022))."

As for the suggestion to add to the discussion, we have added more details on this to the Summary, which now reads: "This difference may be due to chromophores that were insoluble in the solvents utilized (water and methanol) and these insoluble chromophores absorb light more strongly at higher wavelengths (e.g., lower AAEs) than soluble species. For the parameters we

used in this closure analysis, the data suggest that methanol insoluble BrC chromophores contributed roughly 65% and 87% to the light absorption at 532 and 664 nm, respectively".

One of the main concerns of this manuscript is that applied method rely on several assumptions and approximation which can create a large uncertainty in estimation. I appreciate that the authors stated most of the uncertainties for example in extrapolating the wavelength-dependent differences. However, I think the authors should state overall uncertainties in estimating all the absorption values. For example, I think there is a large uncertainty in estimation of the conversion factor itself. How that translate to uncertainties in total absorption?

**Response:** We have added uncertainties to Fig. 5a. We decomposed the uncertainty for the optical closure analysis into two part: 1. from BC, 2. from BrC ($Abs_{TS\ BrC}$*K). The uncertainty from BC was estimated to be 40%, same as the measurement uncertainty of SP2. The uncertainty from BrC was derived from TS BrC measurement uncertainty and the uncertainty for conversion factor K. Error bars representing uncertainties have been added to Fig. 5a and the Figure caption has been edited (see below).

[Figure]

Figure 5. Various light absorption coefficients for the average of the first transect made closest to the Williams Flats fire (23:34–23:39 7 Aug. 2019 UTC). (a) Spectral light absorption closure analysis, where the dashed black line is the light absorption of bare rBC and the solid line is BC considering the coating effect ($E_\lambda$). The brown shading is soluble BrC, $b_{ap,TSBrC,\lambda}$, where $Abs_\lambda^{LWCC}$ was multiplied by the conversion factor $K_\lambda$ to convert from solution to aerosol particle absorption. The upper part of the brown curve is $b_{ap,predicted,\lambda}$, given by Eqn 4. Uncertainties at two extreme wavelengths (300 nm and 700 nm) for two individual components (1. The effect or rBC coating, $b_{ap,BC,\lambda}$, dark gray in plot (a), estimated to be 40%, the same as the SP2 measurement uncertainty; 2. TS BrC $b_{ap,TSBrC,\lambda}$, brown, from ($Abs_{TS\ BrC}$*K), considering the uncertainty in measurements and K). (b) Comparison between $b_{ap,TSBrC,\lambda}$ (brown shading in plot (a)) and $b_{ap,PASBrC,\lambda}$ (difference between red and the black solid line in plot (a)), color coded by wavelength. (c) Similar to plot (b), but versus wavelength (i.e., the difference between BrC determined from the soluble measurements with the conversion factor $K_\lambda$ included, and BrC calculated from the PAS data).

Page 12: Some more discussion about the absorption Angstrom exponent (AAE) measured by PAS and WS BrC and MS BrC and in context to previous study would be useful, like lower AAE value reported by PAS. I'm bit confused with the AAE values from PAS and from TS reported in Figure 3. And how did the authors calculate the modified combustion efficiency?

**Response:** We have added text to address both these points. We first note that the AAE values reported in Figure 3 are of BrC, which can be obtained in a straightforward manner from the TS BrC data. In this case we fit the log of the bap TS BrC vs the log of the wavelength to determine AAE from the slope. We used data over the full wavelength range. For the PAS, the data must be first converted to PAS BrC, which is done using equation 2, and then the data at the three wavelengths are used to determine the AAE using the same log transformation method discussed above. This sentence has been added to Page 12 Line 69: "The AAE for soluble BrC (either WS BrC or TS BrC) measured in FIREX-AQ is comparable to the result from the ATom study (Zeng et al., 2020) but lower than the DC3 (Liu et al., 2015)." Higher AAE observed in the DC3 might be because that study focused on convective regions. BrC aerosol associated with convective movement (secondary production) may have different optical properties (higher AAE), but this hypothesis needs further investigation. No studies reported the AAE value for BrC aerosol from the PAS measurements using the same analytic methods, therefore no comparison was made.

Regarding the modified combustion efficiency, Reviewer 1 has suggested we use OC/rBC instead due to higher dynamic range, which we have done, so all references to MCE have been removed.

Size distribution data and black carbon data from SP2 are missing in the manuscript but it is important to have this information in the SI.

**Response:** The SP2 BC size distribution was added, see Supplement Fig. S1

[Figure]

Figure S1. Mass size distribution of rBC measured for a flight. A log normal fit over the valid range of detection is used to infer the rBC mass undetected in the population.

Authors discussed several other factors that may influence the evolution of BrC. I appreciate this discussion. However, some of the supporting data on this are not shown in the manuscript.

**Response:** We assume this refers to section 3.4.5. We did not present these results in the paper in detail to keep the manuscript focused, but we still felt that it was important to document what other factors we investigated, even if we could come to no conclusion on their possible effects. We have not modified the manuscript.

Summary section can be improved by proving general applicability of the closure exercise and overall applicability of this study. Another thing I find it difficult to draw some firm conclusions as this study investigated several fires with different scales, while chasing one large fire with sufficient time would help to decipher some of the key aspect of BrC evolution. I think this is still an important study but something to discuss in the summary section so future work can be better designed.

**Response:** We note that the closure analysis was applied to all the smoke plumes. We showed the detailed spectral comparison in Fig. 5 for one pass through a plume, but did comparisons for all passes through the smoke plumes in Fig. 6. As for overall applicability of this closure exercise to other types of smoke or other studies, we have no insights since this was the first time this type of closure analysis was performed. We did investigate fires of different spatial scales (or age of smoke plumes), but most of the smoke sampled in this study was focused on near the fires. However, we have added some more details on the evolution of more aged smoke (two plots were added to the supplemental material) and we do make suggestions for future work in the last line of the Summary.

Minor comments:

Page 21, line 67: please check the sentence, 405 nm wavelength was mentioned twice

**Response:** This was corrected by removing the second reference to 405 nm in the sentence.

Overall, there are several acronyms and subscripts, it will be difficult for readers to follow, it will be good to have a table with all the acronyms.

**Response:** The following table has been added as Table 2.

Table 2. Nomenclature

| Nomenclature | Description |
| --- | --- |
| $Abs_{x,\lambda}^{LWCC}$ | Light absorption coefficient of component $x$ at wavelength $\lambda$ in liquid solution measured with LWCC |
| $AAE_x$ | Absorption Ångström Exponents of component $x$ determined from a power law fit over the wavelength range of 300 nm to 500nm. |
| $A_\lambda$ | Light absorption measured by the spectrophotometer with the LWCC |
| $b_{ap,x,\lambda}$ | Light absorption coefficient of component $x$ at wavelength $\lambda$ in aerosol phase |

| | |
|---|---|
| $c_{rBC}$ | Mass concentration of rBC measured by the SP2 |
| $E_\lambda$ | Absorption enhancement for BC particle due to coating effect |
| LWCC | Liquid waveguide capillary cell |
| $K_\lambda$ | Conversion factor for estimating the particle light absorption coefficient from the solution |
| $MAC_{x,\lambda}$ | Mass absorption cross-section of component $x$ at wavelength $\lambda$ |
| $NEMR_x$ | Normalized excess mixing ratio of component $x$ |
| PAS BrC | Brown carbon inferred from the PAS and SP2 |
| rBC | Refractory black carbon concentration measured by the SP2 |
| TS (BrC) | Total-soluble brown carbon determined from the sum of sequential water then methanol extraction of a single filter |
| WS (BrC) | Water-soluble brown carbon |
| 4-NC | 4-Nitrocatechol |

Is there a reason to just show one specific fire events for the wavelength dependencies in the main manuscript? I see the value for each fire events but how about combining all the dataset to get a broader picture?

**Response:** This relates to the question above on including an uncertainty/sensitivity analysis. Our approach was to provid a detailed analysis for one plume transect (Fig. 5) and show similar plots in the supp. material for all the transects of the one plume (Fig. S5). We then made scatter plots summarizing the differences for all plumes (Fig 6). We feel that is sufficient detail for this manuscript and as noted above a much more detailed analysis is planned for a future paper.

Figure 4 can be move to SI. Did you calculate the correction factor for each fire events? I suggest adding some sorts of histograms and combining all the events, so reader can get an idea about the spread of the data.

**Response:** We have left Fig 4 in the main text because we think it is important for the reader to realize that the correction factor has a wavelength dependance, which has not been discussed in previous publications. The suggestion to do a more complete sensitivity analysis to determine the range of values for this factor (maybe at some specific set of different wavelengths), is a good idea, but again we feel it is beyond the scope of this paper and better suited to a dedicated publication.

**Other changes to the manuscript:**

Figure 5. An updated Figure 5 was produced where the PAS fit in Fig. 5a was slighted modified (a constant in the fit equation was removed). This results in a very minor change in the red line in Fig. 5a.

Figure 7. An error was found in Figure 7. The PAS fit did not match the same curve shown in Fig 5a. Figure S4 has been corrected, and Figure S5 and the sentence in Page 14 Line 27 has been deleted.

Figure 8. This figure has been updated. We have added the Tucker near fire marker to the legend and corrected the 807 Williams Flats data points, which results in a slight shift to shorter times for a few data points.

We added two figures to the Supplemental material, equivalent to Fig 8, but only showing the RIM and Tucker fires and RIM and Williams Flats fires to help clarify the discussion on the comparisons of these plumes to the RIM fire reported in an earlier study.